EMBO
reports

# The olfactory receptor Olfr78 promotes differentiation of enterochromaffin cells in the mouse colon

Gilles Dinsart [ID][1,2], Morgane Leprovots [ID][1], Anne Lefort[1,3], Frédérick Libert[1,3], Yannick Quesnel [ID][2,4], Alex Veithen[2], Gilbert Vassart[1], Sandra Huysseune[2], Marc Parmentier [ID][1] & Marie-Isabelle Garcia [ID][1✉]

## Abstract

The gastrointestinal epithelium constitutes a chemosensory system for microbiota-derived metabolites such as short-chain fatty acids (SCFA). Here, we investigate the spatial distribution of Olfr78, one of the SCFA receptors, in the mouse intestine and study the transcriptome of colon enteroendocrine cells expressing Olfr78. The receptor is predominantly detected in the enterochromaffin and L subtypes in the proximal and distal colon, respectively. Using the Olfr78-GFP and VilCre/Olfr78flox transgenic mouse lines, we show that loss of epithelial Olfr78 results in impaired enterochromaffin cell differentiation, blocking cells in an undefined secretory lineage state. This is accompanied by a reduced defense response to bacteria in colon crypts and slight dysbiosis. Using organoid cultures, we further show that maintenance of enterochromaffin cells involves activation of the Olfr78 receptor via the SCFA ligand acetate. Taken together, our work provides evidence that Olfr78 contributes to colon homeostasis by promoting enterochromaffin cell differentiation.

Keywords SCFA; Odorant; Enteroendocrine Cells; Serotonin; Organoids
Subject Categories Development; Membranes & Trafficking

## Introduction

The gastrointestinal epithelium is an important contributor to endocrine physiology and metabolic control by constituting a chemosensory system for external toxic agents, as well as diet-derived nutrients or microbiota-derived metabolites. This "sensory" function is mainly exerted by a subset of epithelial cells, known as enteroendocrine cells (EEC). In response to stimuli, these highly specialized cells secrete neurotransmitters or a variety of hormones having paracrine digestive and endocrine functions (Gribble and Reimann, 2019). Recent reports have uncovered the extraordinary complexity of this lineage, composed of multiple subtypes, each producing a defined set of specific hormones/signaling molecules (Basak et al, 2017; Egerod et al, 2012; Haber et al, 2017; Habib et al, 2012; Roth et al, 1992). In addition, EEC subtypes are also differentially distributed along the digestive tract, according to both proximal-to-distal and crypt-villus axes (Beumer et al, 2018; Haber et al, 2017; Roth et al, 1992). Two main EEC subtypes populate the colon, the so-called L cells devoted to secretion of GLP-1 and PYY peptides and the enterochromaffin cell subtype (EC) that secretes most of the serotonin (5-HT for 5-hydroxytryptamine) produced by the body (Gribble and Reimann, 2019).

Till recently, not much was known about the identity of the molecules that sense the chemical stimuli enabling signal transduction in EECs to promote hormone or neurotransmitter secretion. Accumulated evidence revealed the key role of some G protein-coupled receptors (GPCR) in such process. These receptors recognize as natural ligands the luminal short-chain fatty acids (SCFA) generated by the microbiota (Bellono et al, 2017; Pluznick, 2016). The most abundant SCFAs, acetate, propionate, and butyrate, are particularly concentrated in the colon, where they reach the millimolar concentration range (Cong et al, 2022). The FFAR2/GPR43 and FFAR3/GPR41 receptors, expressed in the EEC L subtype, recognize acetate, propionate, butyrate, and isovalerate as ligands (Le Poul et al, 2003). Whether they are involved in hormonal secretion in these cells is still debated (Christiansen et al, 2018; Psichas et al, 2015; Tolhurst et al, 2012). Loss of function studies in mice have demonstrated that the mouse ortholog Ffar2/Gpr43 and Ffar3/Gpr41 receptors participate to intestinal homeostasis by regulating inflammation in response to chemically induced colitis (Kim et al, 2013). The two other SCFA binding molecules are the olfactory receptors OR51E1 and OR51E2 (the human orthologs of the mouse receptors Olfr558 and Olfr78, respectively) (Pluznick et al, 2013; Saito et al, 2009; Audouze et al, 2014). Although odorant receptors are mainly expressed in olfactory sensory neurons of the olfactory epithelium, Olfr78, and Olfr558 expression is also detected in other tissues including the digestive tract (Bellono et al, 2017; Billing et al, 2019; Fleischer et al, 2015; Lund et al, 2018). It has been demonstrated that binding of SCFAs (isovalerate, butyrate) to the mouse Olfr558 receptor in EC

[1]Institut de Recherche Interdisciplinaire en Biologie Humaine et Moléculaire (IRIBHM), Faculty of Medicine, Université Libre de Bruxelles ULB, Route de Lennik 808, 1070 Brussels, Belgium. [2]Chemcom, Route de Lennik 802, 1070 Brussels, Belgium. [3]BRIGHTcore Facility, IRIBHM, Faculty of Medicine, Université Libre de Bruxelles ULB, Route de Lennik 808, 1070 Brussels, Belgium. [4]Present address: Inchinn Therapeutics, Rue Auguste Piccard 48, 6041 Gosselies, Belgium. ✉E-mail: Marie.Garcia@ulb.be

cells of the small intestine activates basolateral 5-HT release from secretory granules and thereby stimulates afferent nerve fibers (Bellono et al, 2017). The Olfr78 receptor (human ortholog OR51E2) that recognizes acetate and propionate as ligands, was firstly detected in the EEC L subtype (Fleischer et al, 2015). Moreover, transcriptome analyses have also reported its expression in EC cells in the colon (Lund et al, 2018). Using the transgenic knockin/knockout Olfr78-GFP/LacZ mouse line, Kotlo et al have demonstrated that absence of Olfr78 expression is associated with higher levels of intestinal inflammation and worse histopathological score as compared to control mice in an experimental model of colitis (Kotlo et al, 2020). However, the role, if any, of Olfr78 in intestinal epithelium under homeostasis remains to be addressed.

In the present work, we investigated the complete expression profile of Olfr78 in gut and uncovered the biological function of this SCFA receptor in mouse colon epithelium using transgenic mouse lines and organoid cultures. Our findings reveal that signaling through the Olfr78 receptor regulates EC lineage differentiation in colon epithelium and participates to tissue homeostasis.

# Results

## SCFA receptors exhibit unique expression profiles along the small intestine and the colon

To fully dissect the distribution pattern of SCFA receptors, namely Olfr78, Olfr558, Ffar2, and Ffar3, along the mouse small and large intestines, we first analyzed their gene expression on mouse control tissues by qRT-PCR experiments. While *Ffar2* and *Ffar3* transcripts were detected at similar levels along the intestine, higher levels of *Olfr78* and *Olfr558* transcripts were found in the colon as compared to the small intestine (Fig. 1A). In situ hybridization studies further showed that *Olfr78*, *Olfr558*, and *Ffar3* were expressed in discrete countable cells in the epithelium (Fig. 1B–D). In the proximal colon, epithelial *Olfr78* and *Olfr558*-expressing cells were mostly located at the crypt base, while *Ffar3*-positive ([+ve]) cells were mainly found at the top of the crypts, near the lumen, suggesting that Olfrs and Ffar3 receptors may not be co-expressed in the same cell types (Fig. 1B,D). Regarding *Ffar2*, this receptor was more diffusely expressed and showed a decreasing gradient from the bottom to the bottom-half of colon crypts (Fig. 1B). In addition, *Olfr78* and *Olfr558* were expressed in isolated mesenchymal cells and submucosal and myenteric plexuses in the colon (Fig. EV1A,B). Of note, expression of *Olfr558* in the distal colon was only attributed to non-epithelial cells (Figs. 1A–C and EV1B). Finally, in agreement with a previous report (Nøhr et al, 2013), *Ffar3*[+ve] cells were present in the myenteric plexuses (from ileum to distal colon) and *Ffar2* expression was detected in submucosal cells (likely leukocytes) (Fig. EV1C,D). Altogether, these data revealed a unique expression profile of SCFA receptors along the proximal-to-distal and crypt bottom-top villus axes in the gut.

## Olfr78 is expressed in different subtypes of enteroendocrine cells in the colon

Since *Olfr78* expression was particularly abundant in the colon where SCFA ligands are massively produced, we further explored its expression

profile in colon epithelial cells using the knockin-knockout reporter Olfr78[tm1Mom] mouse strain (thereafter referred as Olfr78-GFP). As depicted in Fig. 2A, Olfr78-GFP mice harbor a GFP-IRES-tauLacZ cassette in place of the coding exon of *Olfr78* (Bozza et al, 2009). Using GFP reporter as a surrogate to identify Olfr78[+ve] cells, Epcam[+ve] /GFP[+ve] cells were sorted from the colon of 4 heterozygous (HE) Olfr78[GFP/+] mice (two pools, each obtained from 2 individual mice) by flow cytometry (Figs. 2A and EV2A). All epithelial cells (Epcam[+ve]) were also isolated from a wild-type (WT) colon (Fig. 2A). Following bulk RNA sequencing (seq) of sorted cells, we compared the transcriptome of epithelial Olfr78[+ve] cells to that of all Epcam[+ve] cells with the Degust software to identify significantly up and downregulated genes in Olfr78-expressing cells. Using a cut-off of false discovery rate (FDR) < 0.001 and a $\log_2$ fold-change of >2 or <−2, we identified 905 enriched genes and 321 de-enriched genes (Fig. 2B). As expected, *Olfr78* expression was 60-fold enriched in GFP[+ve] cells (Fig. 2B,C). Regarding other SCFA receptors, *Olfr558* and *Ffar2* were found enriched by 50- and 4-fold, respectively, whereas *Ffar3* expression was not detected at significant levels in GFP[+ve] cells (Fig. 2B,C). High levels of expression of general EEC (*Chga*, *Syp*), EC cells (*Chgb*, *Tph1*, *Tac1* and *Ddc*) and L cells (*Gcg*, *Pyy*, *Insl5*) marker genes were found upregulated (Fig. 2C). Of note, neural markers (*Tnr*, *Nrxn1*, *Hap1*, *Lrrn1*, *Ntm*) were observed in Olfr78-expressing cells, pointing out their neuroendocrine identity. GFP[+ve] cells were especially enriched in presynaptic and postsynaptic markers (*Syt14*, *Snap25*, *Syn1* and *Nlgn2*, *Dlg3*, *Shank2*, respectively) and expressed the neurofilament marker *Nefm* (Fig. 2D). These data are coherent with the function of EC cells, reported to establish interactions through neuropod processes with surrounding epithelial cells and synapses with enteric nerve cells (Bellono et al, 2017; Bohórquez et al, 2014, 2015). Moreover, since EEC differentiation involves transient intermediate committed states, each coined by a particular set of transcription factors (Gehart et al, 2019), we investigated the expression of these genes in Olfr78-expressing cells (Fig. 2E). GFP[+ve] cells did not express early EEC markers (*Dll1*, *Isx*) but were enriched in key early/ intermediate and intermediate transcriptions factors (TF) (*Pbx1*, *Rybp*, *Sox4* and *Pax4*, *Rcor2*, *Rfx3*, respectively), indicating that Olfr78 transcription initiates in these EEC progenitors. Expression of *Olfr78* was also maintained in intermediate/late and late EEC cells based on enrichment of GFP[+ve] cells in the following TFs (*Insm1*, *Neurod1*, *Rfx6*, *Lmx1a* as well as *Pax6*, *Egr3*, and *Emb*) (Figs. 2E and EV2B,C). Altogether, these data indicated that Olfr78 is expressed in epithelial EEC progenitors during their commitment towards L and EC lineages and in mature EEC subtypes.

We confirmed these results on colon tissues of Olfr78-GFP HE mice by immunofluorescence studies. Indeed, the GFP reporter was detected in the EC and L cell lineages (in accordance with 5-HT and PYY expression, respectively) (Fig. 2F,G). Interestingly, GFP was expressed in L (PYY[+ve]) cells exclusively located in the distal colon, in agreement with a previous report (Billing et al, 2019). In contrast, GFP expression in EC (5-HT[+ve]) cells was predominant in the proximal colon (Fig. 2F,G). Overall, these results revealed a regional expression pattern of colon epithelial Olfr78-expressing cells in EEC subtypes.

## Loss of Olfr78 impairs terminal differentiation into enterochromaffin cells

To investigate whether Olfr78 could play any role in the colon, we took advantage of the Olfr78-GFP knockin/knockout mouse line.

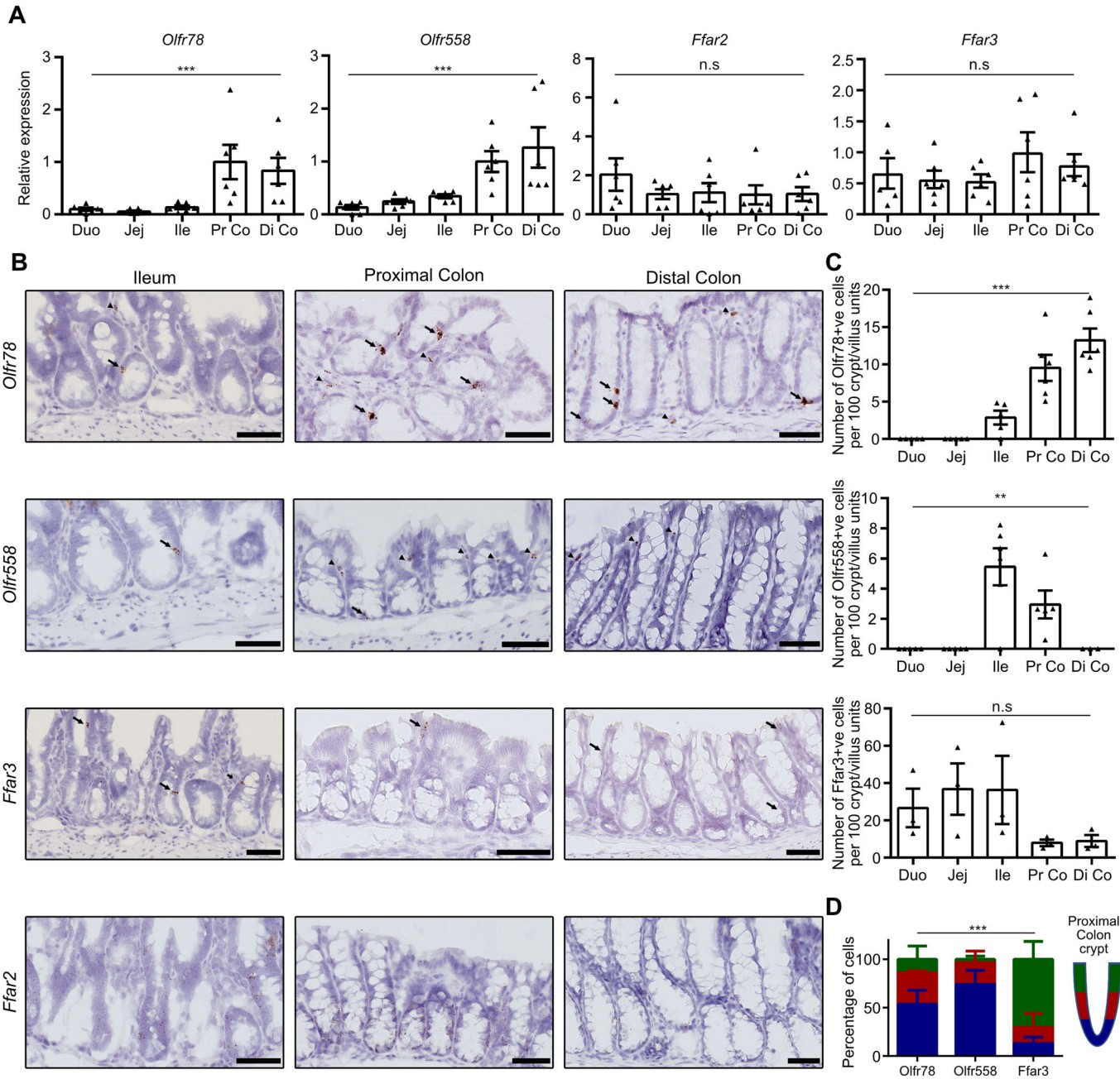

**Figure 1. SCFA receptors exhibit unique expression profiles along the small intestine and colon.**

(**A**) The expression profile of SCFA receptors was analyzed by qRT-PCR on wild-type (WT) mouse intestinal biopsies from duodenum (Duo), jejunum (Jej), ileum (Ile), proximal colon (Pr Co) and distal colon (Di Co). Relative expression levels were arbitrary set to 1 in Pr Co samples. Each symbol indicates the value for a given mouse ($n = 6$). (**B**) The expression profile of SCFA receptors was analyzed on WT mouse tissues by RNAscope. Arrows and arrowheads point to epithelial and non-epithelial expressing cells, respectively. (**C**) Quantification of the number of cells expressing the Olfr78, Olfr558 and Ffar3 SCFA receptors per 100 crypt/villus sections from pictures obtained with RNAscope. Each symbol indicates the value of a given mouse ($n = 3$ to 6). (**D**) Distribution of Olfr78, Olfr558, and Ffar3 positive cells along proximal colon crypts (blue: bottom, red: middle, green: top of the crypt, $n = 3$ mice). Data information: Scale bars = 50 μm (**B**). Data are represented as mean ± SEM. (**A**) Kruskal–Wallis tests: ***$P = 0.002$ (Olfr78), ***$P = 0.002$ (Olfr558) and n.s = not significant (Ffar3) with Dunn's multiple comparison tests: *$P = 0.0370$ Duo vs Pr Co and Duo vs Di Co, **$P = 0.045$ Jej vs Pr Co and Jej vs Di Co (Olfr78); **$P = 0.027$ Duo vs Pr Co, **$P = 0.0016$ Duo vs Di Co, *$P = 0.0351$ Jej vs Di Co (Olfr558). (**C**) Kruskal–Wallis tests: ***$P = 0.001$ (Olfr78), **$P = 0.0013$ (Olfr558) and n.s = not significant (Ffar3) with Dunn's multiple comparison tests: *$P = 0.0370$ Duo vs Pr Co and Duo vs Di Co, **$P = 0.045$ Jej vs Pr Co and Jej vs Di Co (Olfr78); **$P = 0.027$ Duo vs Pr Co, **$P = 0.0016$ Duo vs Di Co, *$P = 0.0351$ Jej vs Di Co (Olfr558). (**D**) Two-way ANOVA test (interaction ***$p = 0.008$) with Tukey's multiple comparison tests: bottom crypts: **$p = 0.0067$ for Olfr558 vs Ffar3; top crypts: *$p = 0.017$ for Olfr78 vs Ffar3 and **$p = 0.0031$ for Olfr558 vs Ffar3. Source data are available online for this figure.

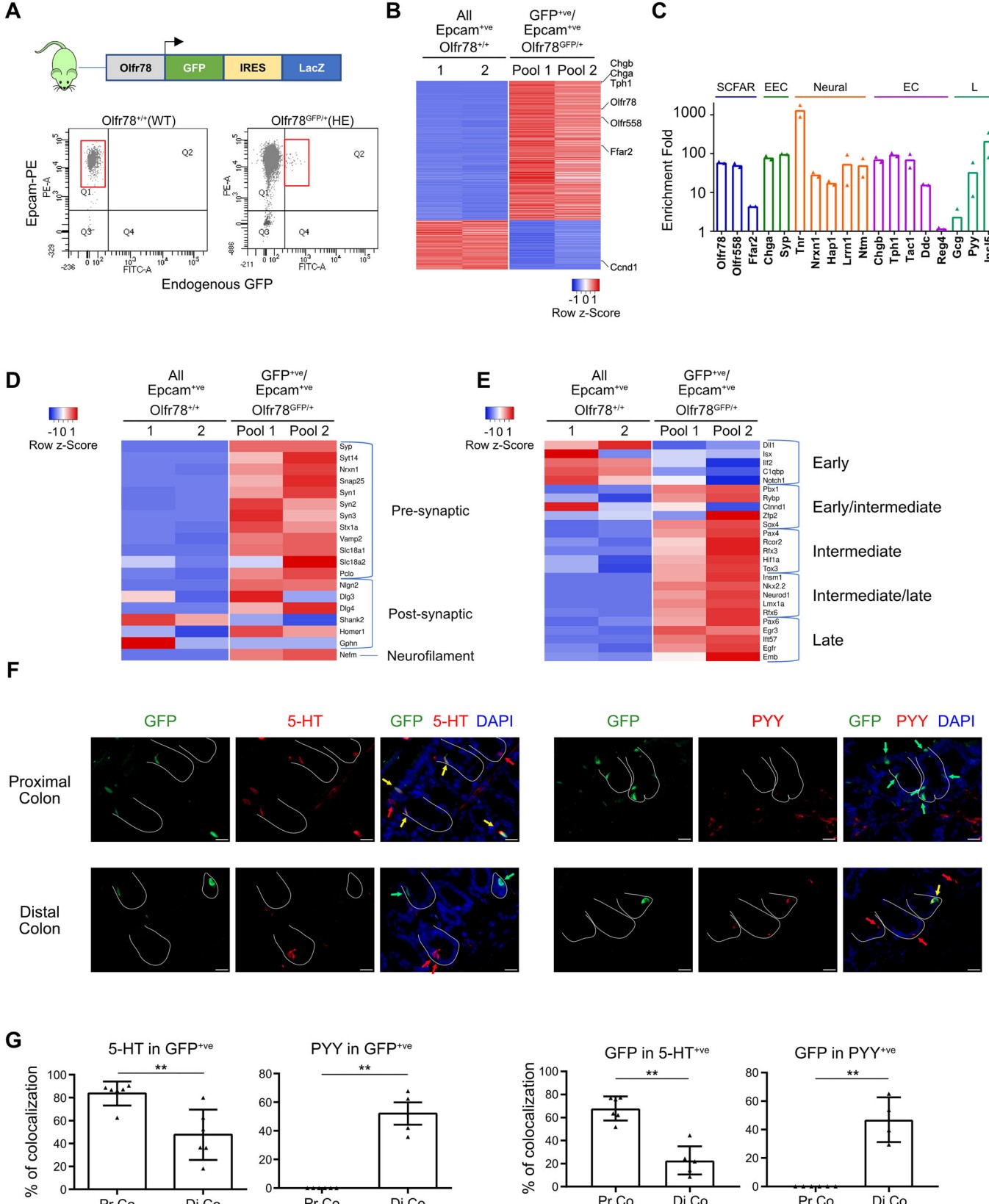

Figure 2. Olfr78 is expressed in different subtypes of enteroendocrine cells in the colon.

(A) Top: Schematic representation of the Olfr78-GFP knockin/knockout mouse line. Bottom: strategy of isolation and sorting by flow cytometry of Epcam$^{+ve}$/GFP$^{+ve}$ cells. Graphs show 10,000 cells and 90,000 cells in Olfr78$^{+/+}$ (WT) and Olfr78$^{GFP/+}$ (heterozygous, HE) samples, respectively. (B) Heatmap of differentially regulated genes in colon Epcam$^{+ve}$/GFP$^{+ve}$ cells (each pool obtained from 2 Olfr78$^{GFP/+}$ mice) versus Epcam$^{+ve}$ cells (2 biological replicates from a single WT mouse) showing log$_2$ fold change < −2 or >2 and false discovery rate ≤ 0.001. (C) Histogram showing fold change of expression of relevant marker genes in Epcam$^{+ve}$/GFP$^{+ve}$ cells vs Epcam$^{+ve}$ cells. (D) Heatmap of differentially regulated pre-synaptic, synaptic, and neuropod-associated genes in Epcam$^{+ve}$/GFP$^{+ve}$ cells vs Epcam$^{+ve}$ cells. (E) Heatmap of differentially expressed transcription factors involved in EEC differentiation in Epcam$^{+ve}$/GFP$^{+ve}$ cells vs Epcam$^{+ve}$ cells. (F) Immunofluorescence showing 5-HT or PYY production in GFP$^{+ve}$ cells in proximal and distal colon of Olfr78$^{GFP/+}$ mice. Crypts are delineated in white. Red and green arrows point to cells expressing only hormones or GFP, respectively. Yellow arrows indicate GFP$^{+ve}$ cells colocalizing with 5-HT or PYY. Nuclei were counterstained with DAPI. (G) Quantification of GFP$^{+ve}$ cells expressing 5-HT or PYY in proximal and distal colon (Pr Co and Di Co, respectively) or 5-HT$^{+ve}$ and PYY$^{+ve}$ cells expressing GFP. Each symbol indicates the value of a given mouse. Data information: Scale bars = 20 µm (F). Data are represented as mean ± SEM. **$P$ < 0.01, Mann–Whitney test (G). Source data are available online for this figure.

First, we confirmed that Olfr78-GFP homozygous mice were knockouts (KO) in the whole colon by qRT-PCR (using primers targeting the coding exon) and by RNAscope (Figs. 3A and EV3A). At the histological level, loss of Olfr78 did not significantly affect Goblet cell differentiation or cell proliferation in colon (Figs. 3B and EV3B). After having checked that GFP$^{+ve}$ cells were detectable in the colon of Olfr78-GFP KO (Fig. EV3C), we sorted by flow cytometry colon Epcam$^{+ve}$/GFP$^{+ve}$ cells from these mice (2 pools, each obtained from 2 individual mice) (Fig. 3C). Then, we compared their transcriptome obtained by bulk RNAseq to that of Epcam$^{+ve}$/GFP$^{+ve}$ cells from Olfr78-GFP HE mice (Fig. 3D, upper panel). This resulted in a list of 364 genes up- and 1,174 genes downregulated in epithelial GFP KO vs HE cells. Strikingly, Olfr78 deficiency correlated with de-enrichment in biological processes associated with neurogenesis, regulation of secretion, and synapse organization (downregulated pre-synaptic and post-synaptic associated genes are listed in Fig. EV3D) meanwhile cell division, DNA replication, and cellular response to stress were upregulated in GFP$^{+ve}$ KOs vs HEs (Fig. 3D, lower panel). Next, focusing on the EEC subtypes previously identified as expressing the Olfr78 receptor, we observed significant two-fold reduction in EC markers involved in serotonin production and metabolism (*Tph1, Ddc, Gch1*), granule secretion (*Chgb, Chga, Rab3c, Gstt1*), and lineage commitment (*Lmx1a, Fev*) in GFP$^{+ve}$ KO cells (Figs. 3E and EV3E). In contrast, similar comparison on L cell marker genes (*Pyy, Gcg, Insl5, Etv1*) did not evidence any clear differential expression pattern in GFP$^{+ve}$ KO cells (Figs. 3E and EV3E). These data suggested that the loss of Olfr78 expression in EEC precursors was specifically interfering with the EC subtype terminal differentiation. Downregulation of *Chgb* expression, but not *Pyy or Gcg*, was confirmed by qRT-PCR on the whole proximal colon exclusively (Fig. 3F). To our surprise, expression of the cognate SCFA receptor Olfr558, whose gene lies 30 kb distant from the Olfr78 gene on chromosome 7, dropped down to 19% residual levels in Olfr78-deficient tissues (Fig. 3F). Accordingly, the numbers of 5-HT$^{+ve}$ and Olfr558$^{+ve}$ cells in proximal colon were significantly reduced (by 26% and 74%, respectively) in the absence of Olfr78 expression (Fig. 3G). Since Olfr558 is expressed by EC cells in the small intestine (Bellono et al, 2017), we further analyzed its expression in colon using a new knock-in Olfr558-BFP line in which BFP is expressed under the control of the Olfr558 promoter (Fig. EV3F). Meanwhile, 60% of Olfr558-expressing cells produced 5-HT, only 35% of EC cells expressed the Olfr558 receptor (Fig. EV3F). In addition, BFP$^{+ve}$ cells did not colocalize with the L cell marker PYY in proximal colon (Fig. EV3F). Next, to fully decipher the identity of epithelial GFP$^{+ve}$ KO cells, we performed single-cell RNAseq

(scRNAseq) on colon cells from Olfr78-GFP WT and Olfr78-GFP KO mice. After data merging using the Seurat package, we isolated and clustered EECs defined by *Chga* expression in epithelial cells (Fig. 3H). In the Olfr78 WT sample, as expected, the EEC cluster was constituted of 2 groups, defined as EC and L cells, based on specific marker genes expression (Fig. 3I,J). In the Olfr78 KO sample, the L cell group was also present and exhibited a transcriptome similar to that of WT L cells. In contrast, the EC group was extremely reduced in Olfr78-deficient tissues and appeared to be replaced by a third cluster of cells in an "undefined state" (Fig. 3I,J). Indeed, this later group of cells expressed low levels of EC marker genes (such as *Tph1, Chgb,* or *Ddc*) and higher levels of non-EEC genes, such as *Sycn* involved in pancreatic acinar cell exocytosis, *Mfsd4a* an atypical solute carrier transporter expressed in several brain areas and *Tfdp2*, a transcription factor involved in cell cycle control (Asle et al, 2005; Chen and Lodish, 2014; Perland et al, 2017). In addition, *Agr2* and *Muc2*, two early goblet cell differentiation markers, were also enriched in Olfr78 KO cells. The observation that *Tph1* was not found downregulated on whole tissues may be explained in part by the fact that this gene, coding for the rate-limiting enzyme in serotonin synthesis, is expressed in about 35% of the EC cells that do not express Olfr78, in few epithelial secretory progenitors in both WT and KO tissues as well as in a fraction of cells from the undefined cluster in Olfr78-deficient mice (Figs. 2G, 3H–J). Altogether, these scRNAseq studies further indicated that loss of Olfr78 in mouse colon results in improper EC differentiation, blocking cells in an undefined state, with characteristics of secretory lineage identity.

## Terminal differentiation into serotonin-producing cells is regulated by epithelial Olfr78 expression

Since Olfr78-expressing cells were detected in epithelium and mesenchyme throughout the colon, we sought to further investigate if the phenotype observed in full Olfr78-GFP KO mice was related to loss of expression in the epithelium, mesenchyme or both compartments. Moreover, this study also aimed at determining whether the double Olfr78/Olfr558 KO phenotype observed in the Olfr78-GFP line could result from unintended Olfr558 KO phenotype due to the proximity of both genes on the genome. For this purpose, we generated a new mouse line harboring an Olfr78 floxed allele targeting the coding exon, thereafter referred to as Olfr78$^{fx}$ (Fig. 4A). To study the impact of epithelial ablation of Olfr78 on EEC differentiation, Olfr78$^{fx}$ mice were crossed with Vil1$^{Cre/+}$ mice (where Cre recombinase is expressed under the control of the Villin promoter, active in epithelial cells) to generate

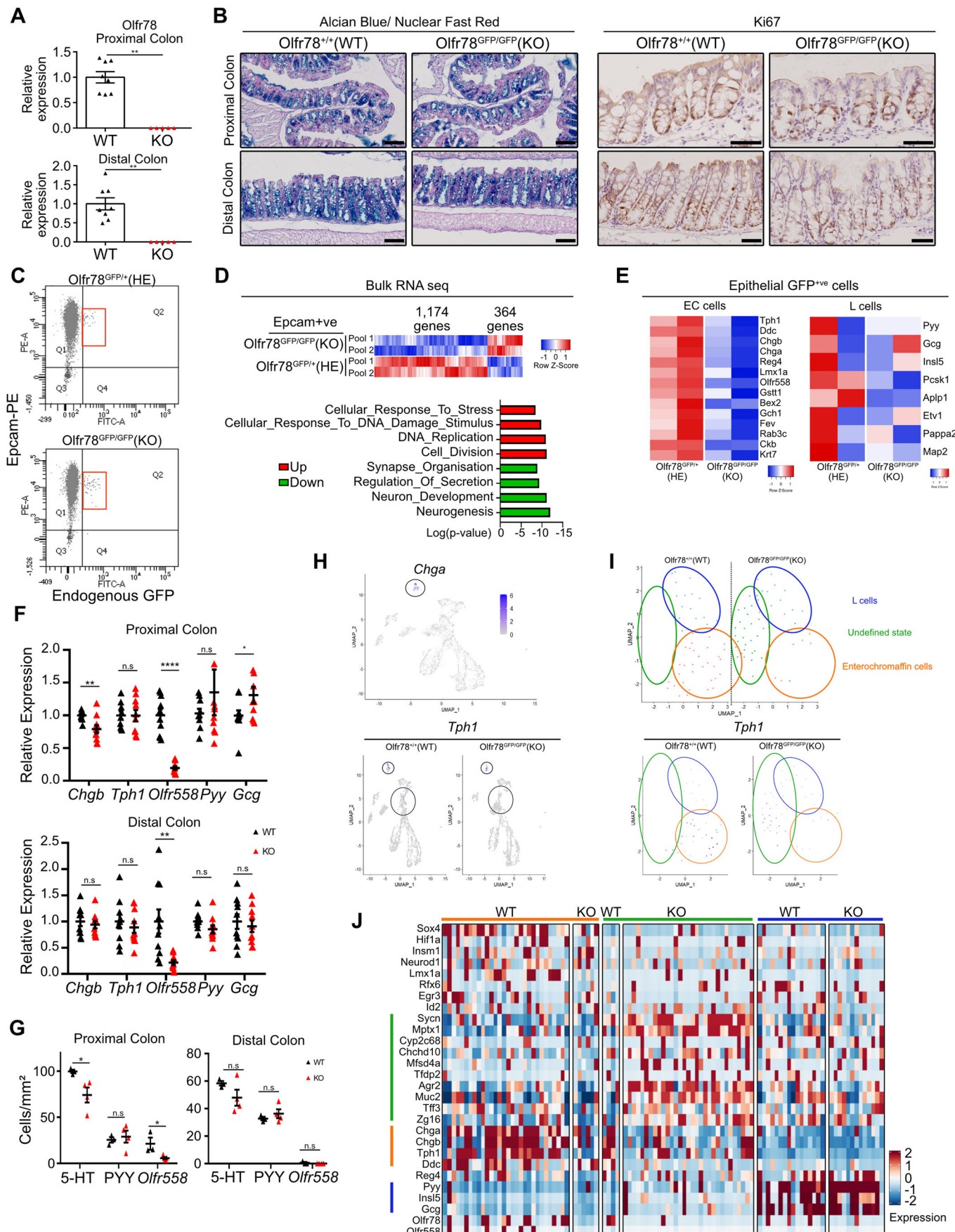

**Figure 3. Loss of Olfr78 impairs terminal differentiation into enterochromaffin cells.**

(A) Analysis of Olfr78 expression by qRT-PCR in proximal and distal colon from WT and Olfr78-GFP KO mice. Relative expression levels were arbitrary set to 1 in WT samples. Each symbol indicates the value of a given mouse ($n = 8$ WT, 5 KO). (B) Representative images showing Alcian Blue-Nuclear Fast Red staining to evidence Goblet cell differentiation (left) and immunohistochemistry for Ki67 staining (right) on proximal and distal colon in Olfr78-GFP WT and Olfr78-GFP KO mice. (C) Bottom: strategy of isolation and sorting by flow cytometry of Epcam$^{+ve}$/GFP$^{+ve}$ cells. Graphs show 90,000 cells in Olfr78$^{GFP/+}$ (HE) and Olfr78$^{GFP/GFP}$ (homozygous/knockout, KO) samples, respectively. (D) Transcriptome comparison between colon epithelial Olfr78-GFP KO vs Olfr78-GFP HE GFP$^{+ve}$ cells by bulk RNA seq. Upper panel: heatmap of differentially expressed genes between epithelial Epcam$^{+ve}$/GFP$^{+ve}$ cells coming from 2 pools of 2 different Olfr78-GFP HE or KO mice. The number of genes differentially modulated is indicated. Lower panel: GSEA-Biological processes for upregulated (red) and downregulated (green) gene lists in Olfr78-GFP KO vs Olfr78-GFP HE GFP$^{+ve}$ cells. (E) Heatmaps showing expression levels of EEC markers in sorted epithelial Olfr78-GFP HE and Olfr78-GFP KO GFP$^{+ve}$ pools. (F) Expression of EEC markers analyzed by qRT-PCR on Olfr78-GFP WT and Olfr78-GFP KO proximal and distal colon biopsies. Relative expression levels were set to 1 in WT samples. Each symbol indicates the value for a given mouse ($n = 10$ WT, 10 KO). (G) Quantification of 5-HT, PYY, and *Olfr558* expressing cells in proximal and distal colon of Olfr78-GFP WT and Olfr78-GFP KO mice performed based on IHC staining (5-HT and PYY) or RNAscope (*Olfr558*). Each symbol indicates the value of a given mouse ($n = 3$ WT, 4 KO). (H) Upper panel: UMAP of merged epithelial cells from WT ($n = 1764$ cells) and KO ($n = 1500$ cells) mice after scRNAseq. The cell cluster showing *Chga* expression is circled. Lower panel: Umap of WT and KO epithelial cells showing *Tph1* expression. (I) Upper panel: UMAPs of WT and KO *Chga*-expressing EEC cells showing clustering into 3 cell populations. Lower panel: Umap of WT and KO EEC cells showing *Tph1* expression. (J) Heatmap showing the expression of various genes in the 3 EEC-associated groups identified in (I). Data information: Scale bars = 100 μm (Alcian Blue) or 50 μm (Ki67) (B). Data are represented as mean ± SEM. (A) Mann–Whitney tests: **$P = 0.0016$ (Pr Co and Di Co). (F) Unpaired t-tests (Tph1 in Pr Co; Chgb, Tph1, and Pyy in Di Co), unpaired t-tests with Welch's correction (Chgb, and Olfr558 in Pr Co and Olfr558 in Di Co), Mann–Whitney test (Pyy in Pr Co). n.s = not significant; *$P < 0.05$; **$P < 0.01$; ***$P < 0.001$. (G) Unpaired t-tests (5-HT and PYY in Pr Co), Mann–Whitney test (Olfr558 in Pr Co and 5-HT, PYY and Olfr558 in Di Co). n.s = not significant; *$P < 0.05$. Source data are available online for this figure.

Vil1$^{Cre/+}$-Olfr78$^{fx/fx}$ mice, referred to as Olfr78 eKO (eKO, for KO in epithelium). First, we validated the efficient deletion of the targeted region on DNA isolated from colon biopsies of Vil1$^{Cre/+}$-Olfr78$^{fx/fx}$ (Fig. EV4A). We also confirmed by RNAscope that loss of Olfr78 was restricted to the epithelium, with non-epithelial expression being preserved and representing 12% residual *Olfr78* expression in proximal colon by qRT-PCR analysis (Fig. 4B,C). Then, we investigated colon histology and mucus production by Alcian Blue staining and did not notice any significant altered Goblet cell differentiation or change in epithelial cell proliferation (Figs. 4D and EV4B). Next, EEC cell differentiation was studied by qRT-PCR using EC and L cell marker genes. As observed in Olfr78-GFP KO mice, Olfr78 eKO mice exhibited a consistent reduction in the EC markers *Chgb* and *Olfr558* (by 35% and 48%, respectively) in proximal, but not distal colon (Fig. 4E). Instead, expression levels of the L cell markers *Pyy* and *Gcg* were not different between the genotypes (Fig. 4E). Moreover, in agreement with qRT-PCR analyses, the density in EC cells was decreased by 32% in the proximal colon of eKO mice as compared to littermate controls whereas no change in EC and L cells density was observed in distal colon (Fig. 4F). Altogether, these findings further demonstrated that epithelial expression of Olfr78 in EECs is necessary to generate mature EC cells in proximal colon. To determine whether Olfr78 loss could alter colon 5-HT levels, we measured this hormone in stools collected from controls and eKO mice. Consistent with overall decrease in EC density in the absence of this SCFA receptor, 5-HT concentration was found tendentially reduced by 25% in eKO mice (p-value = 0.0797) (Fig. 4G). However, this did not affect global stool transit, a process promoted by 5-HT (Fig. EV4C).

## Enterochromaffin cell differentiation involves activation of the Olfr78 receptor via the SCFA ligand acetate

Finally, to investigate the molecular mechanisms by which Olfr78 can contribute to EC differentiation, we used a 3D organoid culture model (Sato et al, 2011). Following 7 days of culture after replating, Olfr78-GFP WT or Olfr78-GFP KO colon organoid lines (generated from individual mice) were stimulated for 48 hs with natural Olfr78 agonists: acetate or propionate at a concentration of

20 mM and 10 mM, respectively (Figs. 5A and EV4D, E). As shown in Fig. 5B, WT organoids appeared more prone to express EEC markers as compared to KOs under untreated conditions. Expression of *Chgb* and *Tph1*, used as the most specific markers of EC maturation, but not the L cell marker *Pyy*, was upregulated by 55% in WT organoids upon acetate challenge, showing that this SCFA stimulates EC differentiation. In contrast, expression of *Chgb* was downregulated by 23% in Olfr78-GFP KOs following acetate challenge as compared to untreated conditions (Fig. 5B). Moreover, propionate induced downregulation of EC and L cell markers irrespective of the genotype (Fig. EV4D). Note that expression of *Olfr558* and *Gcg* (markers of EC and L cells, respectively) remained at very low levels whatever the tested organoids and culture conditions (Fig. EV4E). Together, these data indicated that maintenance of EC maturation involves activation of the epithelial Olfr78 receptor via its ligand acetate in the mouse colon.

## Loss of Olfr78 expression alters colon homeostasis

Having provided evidence that epithelial Olfr78 regulates the production of functional EC cells known to contribute to colon physiology, we investigated whether absence of Olfr78 could affect global colon homeostasis. First, we analyzed the whole epithelium by studying the transcriptome of isolated crypts from Olfr78-GFP and Vil1$^{Cre/+}$-Olfr78$^{fx}$ mice ($n = 5$ WT/controls and 7 KO/eKOs) by bulk RNA seq. Analysis of differentially expressed genes (FDR ≤ 0.2 and log2 fold-change of 0.585) resulted in a list of 35 up- and 40 downregulated genes in Olfr78-deficient vs control crypts (Fig. 6A). MDS plots showed separate clustering of both genotypes (Fig. 6B). In silico studies on modulated biological processes revealed de-enrichment in genes involved in defense response to bacteria, interspecies interaction between organisms and pyroptosis (Fig. 6C). Analysis of colon epithelial (Epcam$^{+ve}$) cells by scRNAseq revealed complete loss of two Goblet-like populations (clusters 6 and 8) characterized by the expression of *Reg4*, *Mpst*, *Serum amyloid A1*, and *Papss2* genes (Figs. 6D and EV5A). In addition, concomitant amplification of secretory progenitors (cluster 0) was detected in Olfr78-deficient tissues, suggesting that absence of Olfr78 expression had altered cell fate in the epithelium with an

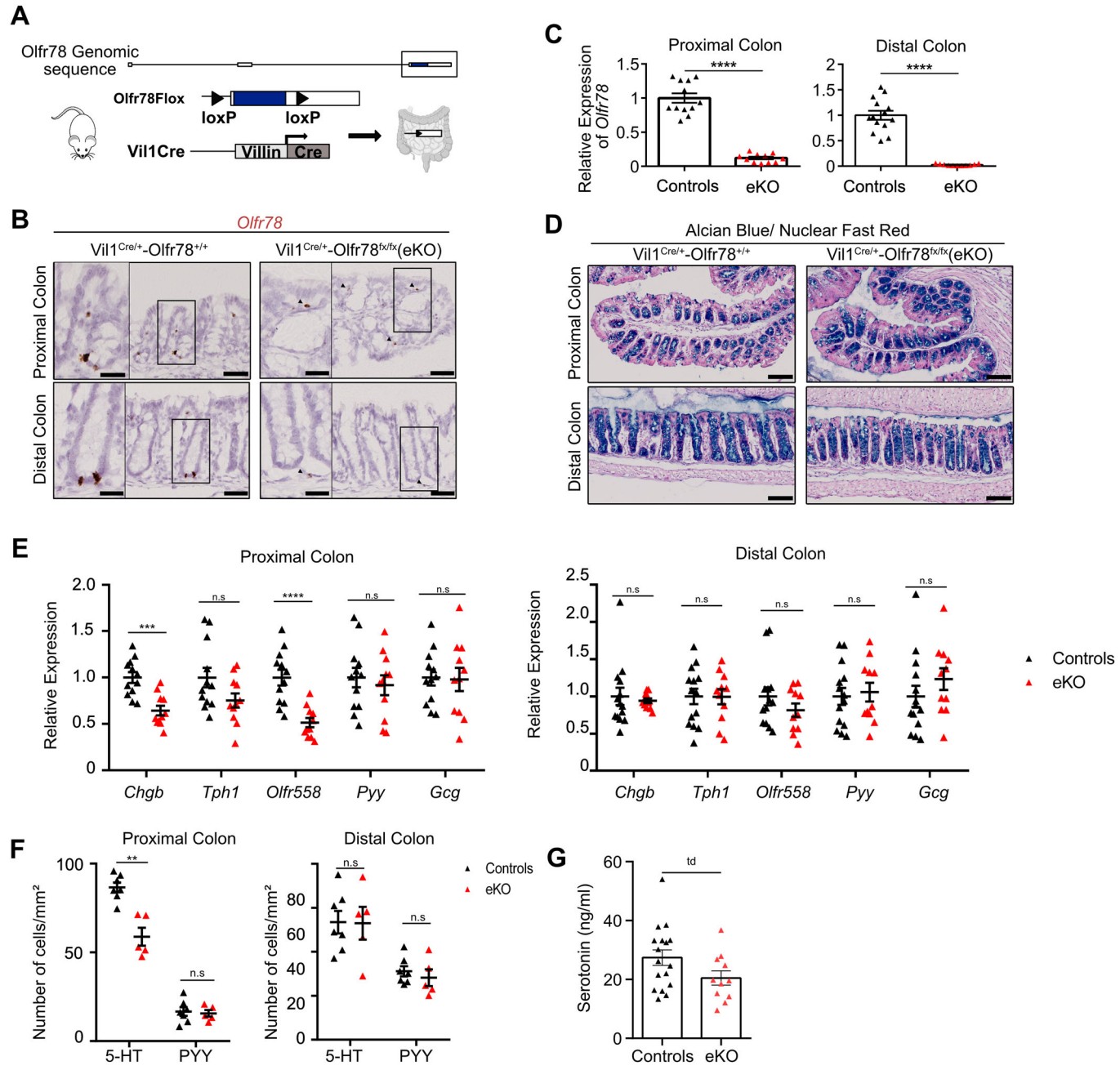

**Figure 4. Terminal differentiation into serotonin-producing cells is regulated by epithelial Olfr78 expression.**

(A) Genomic construction of the Olfr78$^{fx}$ line. The LoxP sites flanking the exon 3 coding for Olfr78 are evidenced. The coding sequence is labeled in blue. (B) Representative pictures of RNAscope staining showing specific epithelial ablation of *Olfr78* expression in Vil1$^{Cre/+}$-Olfr78$^{fx/fx}$ (eKO) mice but not in Vil1$^{Cre/+}$-Olfr78$^{+/+}$ mice. Arrowheads show non-epithelial expression of *Olfr78*. (C) Analysis of residual *Olfr78* expression by qRT-PCR in control or eKO colon biopsies. Relative expression levels were arbitrary set to 1 in control samples. Each symbol indicates the value of a given mouse ($n = 12$ controls and 11 eKO). Controls corresponded to Vil1$^{+/+}$-Olfr78$^{fx/fx}$ and Vil1$^{Cre/+}$-Olfr78$^{+/+}$ mice. (D) Representative pictures of Alcian Blue-Nuclear Fast Red staining on proximal and distal colon from Vil1$^{Cre/+}$-Olfr78$^{+/+}$ or Vil1$^{Cre/+}$-Olfr78$^{fx/fx}$. (E) Expression of EEC markers was analyzed by qRT-PCR on colon biopsies from controls or eKO mice. Relative expression levels were arbitrary set to 1 in controls. Each symbol indicates the value for a given mouse ($n = 12$–14 controls, 11 eKO). (F) Quantification of 5-HT and PYY-expressing cells in proximal and distal colon tissues from controls or eKO mice performed based on IHC staining. Each symbol indicates the value of a given mouse ($n = 7$ controls, 5 eKO). (G) Serotonin levels in stools collected from controls and eKO colon (mice age: between 8 and 27 weeks old). Each symbol indicates the value of a given mouse ($n = 17$ controls, 11 eKO). Data information: Scale bars: 50 µm (B, large view) or 25 µm (B, inset), 100 µm (D), Data are represented as mean ± SEM. (C) Unpaired t-tests with Welch's correction, ****$P < 0.0001$. (E) Unpaired t-tests (Pr Co), Mann–Whitney test (Di Co). n.s = not significant; ***$P < 0.001$; ****$P < 0.0001$. (F) Mann–Whitney tests, n.s = not significant, **$P < 0.01$. (G) Unpaired t-test, td tendency ($p = 0.0797$). Source data are available online for this figure.

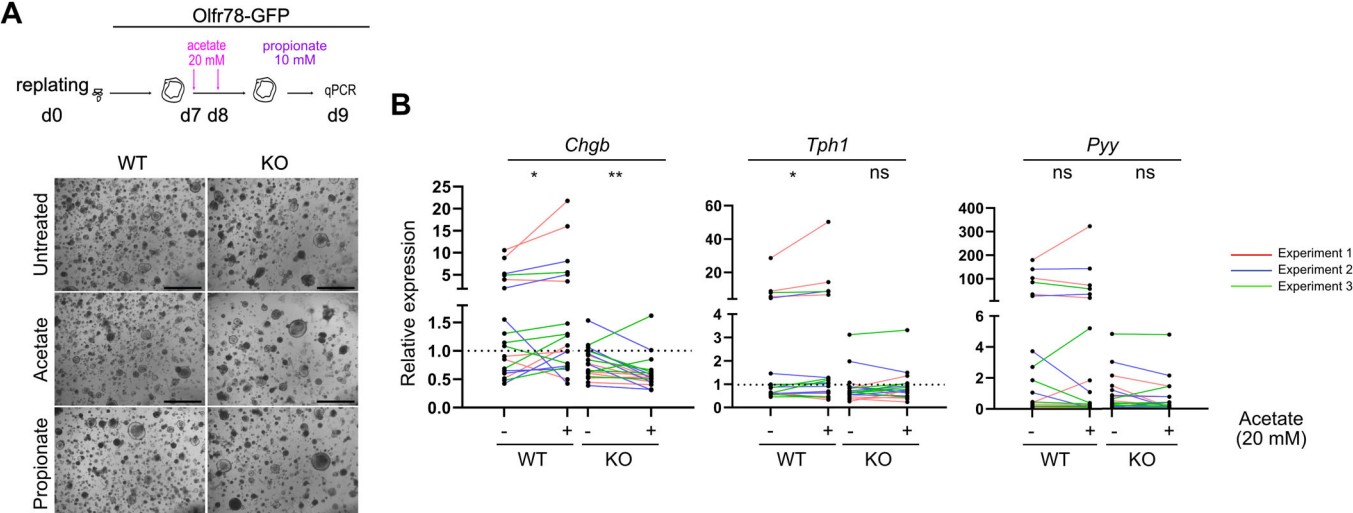

**Figure 5. Enterochromaffin cell differentiation involves activation of the Olfr78 receptor via the SCFA ligand acetate.**

(A) Design of the experiment and representative pictures of Olfr78-GFP WT and Olfr78-GFP KO colon organoid cultures after 48 h of treatment (acetate 20 mM or propionate 10 mM) or untreated conditions. (B) Expression levels of EEC markers analyzed by qRT-PCR on colon organoids after 48 h of treatment. Data are reported as the relative expression levels in treated and control conditions in WT and KO organoids. Each symbol indicates the individual value of a given organoid line in each experiment (n = 6 WT and 6 KO). Colored lines identify paired samples in the 3 independent experiments. Data information: Scale bars: 1 mm (A). Data are represented as mean ± SEM. (B) Wilcoxon matched-pairs signed rank test, n.s = not significant, *P < 0.05, **P < 0.01. Source data are available online for this figure.

increased proportion of cells being engaged into a secretory progenitor fate (Fig. 6D). The significant increase in density of GFP$^{+ve}$ cells observed in Olfr78-GFP KO vs Olfr78-GFP HE colons, secretory in their identity, further sustained this idea (Fig. 6E). Regarding the surrounding non-epithelial cells present in colon, analysis of scRNAseq data from Olfr78-GFP WT and Olfr78-GFP KO mice indicated that they were clustered into 12 distinct populations in both genotypes, without major quantitative differences detected in immune CD45$^{+ve}$ cells or stromal Pdgfra$^{+ve}$ cells in the absence of Olfr78 (Fig. EV5B,C). Next, since 5-HT production and its luminal release is associated with microbiome homeostasis, we studied the impact of Olfr78 loss on fecal microbiota by performing a metagenome analysis on colon stools from Olfr78-GFP and Vil1$^{Cre/+}$-Olfr78$^{fx}$ (n = 18 WT/controls and 15 KO/eKOs) (Fig. 6F,G). First, global phylum analysis revealed differences between the two transgenic lines with increased proportion of *Proteobacteria* at the expense of *Bacteroidetes* and detection of *Deferribacteres* in the Vil1$^{Cre/+}$-Olfr78$^{fx}$ line when compared to the Olfr78-GFP one. Secondly, in the Olfr78-GFP line in which *Bacteroidetes* is the main phylum, the *Firmicutes/Bacteroidetes* ratio, proposed as marker of dysbiosis when dysregulated, was significantly increased in Olfr78-GFP KO vs WT samples, despite no significant difference in total body weight between genotypes (Figs. 6G and EV5D). *Eubacterium* and *Clostridium*, genera belonging to the *Firmicutes* phylum, were over-represented in Olfr78-deficient mice. Conversely, *Turicibacter sanguinis*, a species reported to take advantage of luminal 5-HT released by ECs to increase its fitness and colonization ability in the colon (Fung et al, 2019), appeared virtually absent in Olfr78-GFP KO samples (Figs. 6G and EV5E). No significant difference in the *Firmicutes/Bacteroidetes* ratio was detected between genotypes in Vil1$^{Cre/+}$-Olfr78$^{fx}$ mice; though *Alistipes*, a genus associated with

dysbiosis and the beneficial *Muribaculum intestinale* were reduced in eKO stools (Parker et al, 2020; Smith et al, 2019). In addition, *Akkermancia muciniphila*, a proposed probiotic was over-represented in some eKO samples (Fig. 6G). Knowing that microbiota are the main producers of SCFAs in the colon and having found that Olfr78-deficient mice had signs of dysbiosis, we analyzed the concentration of various SCFA compounds (from C2 to C6) in the stools of mice from the two transgenic lines. No significant differences were detected between genotypes for any of the SCFAs analyzed, including acetate and propionate, the two reported ligands of Olfr78 (Fig. EV5F). In summary, our results indicate that the loss of Olfr78 receptor alters colon homeostasis, characterized by deficient epithelial defense against bacteria and slight dysbiosis under chow diet.

# Discussion

Since their discovery in the olfactory epithelium in 1991 (Buck and Axel, 1991), several olfactory receptors have been reported to be ectopically expressed (Lee et al, 2019). Among these receptors, Olfr78/OR51E2 expression was detected in the gut at significant levels (Billing et al, 2019; Fleischer et al, 2015; Kotlo et al, 2020; Lund et al, 2018). In the present study, we have investigated the biological function of Olfr78 in the colon and revealed that this receptor regulates enterochromaffin cell maturation under homeostatic conditions.

In line with previous studies, using the reporter Olfr78-GFP mouse line, transcriptomic and immunostaining methods, our work confirmed that Olfr78 is mainly present in two epithelial enteroendocrine cell subtypes in colon: EC and L cells, characterized by the expression of the marker genes *Tph1*, *Ddc*, *Tac1* and

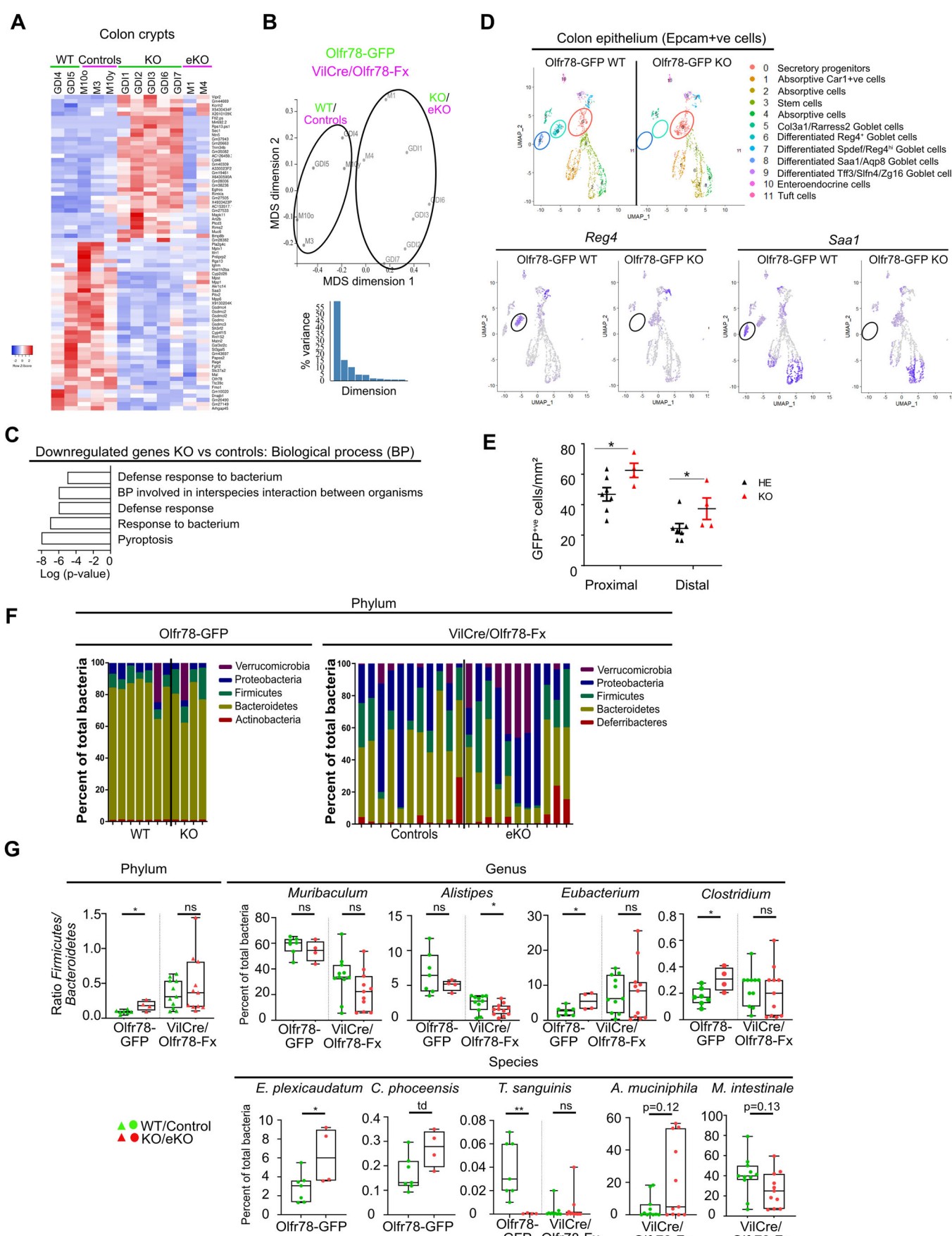

◄ **Figure 6. Loss of Olfr78 expression alters colon homeostasis.**

(A) Left: Heatmap of differentially expressed genes identified by RNAseq on colon crypts isolated from Olfr78-GFP WT and KO mice and Vil1^Cre/Olfr78^fx controls and eKO mice (n = 2WT, 3 Controls, 5 KO and 2 eKO mice) (FDR ≤ 0.2, Log₂FC <−0.585 or >0.585 from Degust). (B) Multi-dimensional scaling (MDS) plot of RNA-seq datasets on genes identified in (A). (C) Modulated Mol-Sig GSEA Biological processes in the transcriptome of WT/Controls vs KO/eKO crypts. (D) Upper panel: Umap of WT and KO epithelial cells showing 12 different clusters. Lower panel: Umap of WT and KO epithelial cells showing *Reg4 or Saa1* expression. Circles identify differentially enriched clusters present in WT and KO colon. (E) Quantification of GFP^+ve cell density in proximal (Pr Co) and distal colon (Di Co) from Olfr78-GFP HE and Olfr78-GFP KO mice performed based on IHC staining. Each symbol indicates the value of a given mouse (n = 7 HE, 4 KO). (F) Histograms showing the relative microbial abundance at the Phylum and Genus levels in Olfr78-GFP mice (n = 7 WT and 4 KO) and Vil1^Cre/Olfr78^fx controls and eKO mice (n = 11 controls and 11 eKO) by metagenome sequencing. (G) Phylum: Ratio of the prevalence of *Firmicutes/Bacteroidetes* populations obtained from the data in (F). Genus and Species: Prevalence of genus or species in percentage of total bacteria obtained from the data in (F). Each symbol indicates the value of a given mouse (Olfr78-GFP line: n = 7 WT and 4 KO; VilCre/Olfr78-Fx line: n = 11 controls and 11 eKO). Data information: Data are represented as mean ± SEM (E) and as box and whiskers defining minima to maxima and median line (G). (C) Hypergeometric distribution test of overlapping genes over all genes in the universe, (E) Unpaired t test (G, *Alistipes*) Unpaired t test: *P < 0.05; (all the others in G) Mann–Whitney tests n.s = not significant; td: tendency (p = 0.0727); *P < 0.05; **P < 0.01.

---

*Pyy*, *Gcg*, *Insl5*, respectively (Billing et al, 2019; Lund et al, 2018). Olfr78 expression was detected in the majority in EC cells (80%) of the proximal colon and was identified in both EC and L cells in the distal colon. By using the resource generated by Gehart et al (2019) on time-resolved single-cell transcriptional mapping of enteroendocrine differentiation in the small intestine, we further determined by bulk and single-cell RNA seq methods that expression of Olfr78 starts in early/intermediate EEC precursors (*Rybp*, *Sox4*) and is maintained throughout differentiation in intermediate/late (*Lmx1a*, *Rfx6*) and late (*Pax6*) stages. Analysis of Olfr78-deficient cells revealed that the absence of this receptor leads to markedly reduced EC cell differentiation, characterized by downregulation of the *Lmx1a* and *Fev* transcription factors (Wang et al, 2010) while no significant decrease in the expression of the transcription factor *Rfx6*, promoting differentiation into L cells, was observed (Piccand et al, 2019). More globally, dysregulated EC differentiation in Olfr78-deficient cells leads to downregulation of neuronal genes, as represented by genes encoding pre-synaptic and secretion granule components. In this regard, the human ortholog OR51E2, sharing 93% identity with Olfr78, was recently detected in human pulmonary neuroendocrine cells and reported to activate a neuroendocrine phenotype in prostate cancer cells (Abaffy et al, 2018; Kuo et al, 2022). Taken together with our results, these data suggest that, in response to its ligands, the Olfr78/OR51E2 receptor can promote a neuronal-like phenotype outside the olfactory epithelium. Interestingly, a study recently revealed that SCFA ligands (acetate or propionate), used at concentrations like those found in the colon (10 mM), promote EC cell differentiation in colon organoid cultures, although the identity of the transducing receptor(s) was not investigated (Tsuruta et al, 2016). In the present work, we provide further evidence that EC cell maturation involves, at least in part, the Olfr78 receptor, through the activation by one of its ligands acetate, but not propionate. The observed negative impact of this later agonist on organoids' gene expression may involve epigenetic regulation through HDAC activity inhibition as reported previously (Grouls et al, 2022; Silva et al, 2018). Regarding the L cell subtype, density and expression levels of the anorexigenic PYY peptide in the colon did not appear modified in absence of Olfr78 expression. However, it is not excluded that this receptor could regulate secretion of this hormone in L cells as proposed by Nishida et al (2021).

Despite a substantial altered EC identity detected in Olfr78-deficient cells in both transgenic lines, this resulted in a consistent but modest reduction (by 30%) in the number of serotonin-producing EC cells in the proximal colon and a 25% reduction in fecal 5-HT levels without significant impact on global gut transit. It is likely that the presence of luminal 5-HT produced by the rest of the intestine can locally compensate transit for the EC cell reduction observed in the proximal colon. Moreover, expression of the *Tph1* gene, did not appear downregulated at the whole tissue level. Several hypotheses, not mutually exclusive, could be proposed to explain these observations. Firstly, 35% of EC cells do not express Olfr78 in the proximal colon, and can a priori differentiate normally, being able to produce some 5-HT. Secondly, as reported in the small intestine by Haber et al (2017), some expression of *Tph1* was still detected in early secretory progenitor/EEC precursors and in Chga^+ve cells of the undefined cluster in Olfr78-deficient cells. Therefore, this may contribute to dampen the observed reduction of *Tph1* expression in Olfr78-deficient EC cells. The major impact of Olfr78 loss detected at the whole tissue level was on chromogranin's expression, especially chromogranin B. This later is a component of secretory granules recently reported to form a chloride channel involved in the secretion of neurotransmitters (Yadav et al, 2018). It is expected that the reduction in chromogranin B levels in Olfr78-deficient cells would also alter the 5-HT secretion process.

The present study also sheds light on the complex expression profile of SCFA receptors in colon where the concentration of their ligands is the highest. Indeed, Olfr558 and Ffar2 receptors were found enriched in Olfr78-expressing cells. This suggests different genetic and/or epigenetic regulatory mechanisms of olfactory receptors' expression in EC as compared to olfactory sensory neurons where one neuron expresses only one receptor (Serizawa et al, 2004). Moreover, we unexpectedly found that the loss of Olfr78 induced a decrease in Olfr558 expression in EC cells. Although not formally excluded, the likelihood that this would result from an artifact of the transgenic strategy, affecting the genetic locus that bears the two genes, remains low since the effect was observed in both mouse lines. In addition, expression of both odorant receptors only partially overlaps in the epithelium of proximal colon, in a subfraction of EC, and not at all in L, cells. Regarding the hypothetical function of Olfr558 in colon EC cells, one possibility could be that this receptor exhibits in colon, similar functions as those described in the small intestine (Bellono et al, 2017). Stimulation of Olfr558 with the butyrate and isovalerate ligands also present in colon would promote 5-HT basolateral release from EC cells to activate neuronal circuitries. Further experiments will be needed to investigate this hypothesis. Besides,

our study also revealed that Olfr78-expressing EC cells express the Ffar2 receptor. Contrary to Olfr78 and Olfr558, this receptor was also detected in other cells in the bottom of colon crypts, suggesting that it would exert a more general function in epithelial cells than odorant receptors. Previous studies have demonstrated that Ffar2 regulates inflammation in colon (Kim et al, 2013). However, the potential role of Ffar2 in colon EC cells remains to be determined; especially considering that this receptor, like Olfr78, recognizes acetate and propionate as ligands, but that these two receptors induce signaling through opposite cascades involving Gq/Gi and Gs proteins for Ffar2 and Olfr78, respectively (Le Poul et al, 2003; Saito et al, 2009). Interestingly, it was previously demonstrated that Olfr78 and Ffar3 coordinate their activity to regulate blood pressure (Pluznick et al, 2013), raising the possibility of a similar mechanism involving Olfr78 and Ffar2 in EC cells. Additional studies are needed to explore this hypothesis and to further decipher the complex interplay between both receptors in the regulation of EC cell maturation.

At the global tissue level, despite intrinsic differences observed between the two transgenic lines having distinct genetic backgrounds (Olfr78-GFP and Vil1$^{Cre/+}$-Olfr78$^{fx}$), we detected that Olfr78-deficient mice had a slightly modified microbiota, an observation also made by Kotlo et al (2020). Indeed, Olfr78 KO mice exhibit an increased *Firmicutes/Bacteroidetes* ratio as compared to WT littermates. In addition, despite the modest reduction in luminal 5-HT levels in Olfr78-deficient mice, *Turicibacter sanguinis*, a bacterium reported to need serotonin to increase its fitness, was virtually absent in Olfr78-GFP KO stools, showing that alteration in the epithelium significantly impacted the microbiota (Fung et al, 2019). Since the *Firmicutes/Bacteroidetes* ratio is increased in obesity, it would be interesting to further investigate the impact of Olfr78 loss on the global metabolic status under high-fat diet conditions. In epithelial Olfr78-deficient mice, reduced abundance of the genera *Alistipes* and *Muribaculum* were observed meanwhile *Akkermansia muciniphila* appeared over-

represented in part of the cohort as compared to control mice. Since gut over-colonization by this species is reported to promote excessive mucin degradation and reduce intestinal barrier integrity, over-representation of *A. muciniphila* may contribute to exacerbate sensitivity of eKO mice to any further luminal colon perturbation (Qu et al; 2023). Finally, our transcriptomic data on colon crypts showed that loss of Olfr78 expression modifies colon epithelium homeostasis by reducing defense response to bacteria, but without any major impact on the non-epithelial compartments. Goblet-like populations expressing the *Mpst* and *Papss2* genes, involved in sulfation metabolism and regulating inflammation, were found absent in Olfr78-deficient colon (Zhang et al, 2022; Xu et al, 2021). Of interest, in inflammatory conditions experimentally induced by dextran sodium sulfate treatment, complete absence of Olfr78 expression increases inflammation and impairs efficient tissue regeneration (Kotlo et al, 2020). Knowing that Olfr78 is also expressed in mesenchymal cells in colon, it will be worth addressing the putative role of these cells in tissue homeostasis during regeneration.

In summary, in the present work, we provide evidence that the Olfr78 receptor is expressed in colon enteroendocrine precursors and is required for proper maturation into the EC cell subtype, devoted to serotonin release. Loss of Olfr78 leads to slight dysbiosis and modifies the ability of defense response to bacteria in colon crypts. Given that 5-HT represents a major potential pharmacological target in metabolic disorders and inflammatory bowel diseases, further exploration of the role of Olfr78 in epithelial and stromal compartments under pathological conditions will help to fully elucidate the complex mechanisms regulating SCFA receptors, 5-HT secretion, and colon homeostasis.

## Methods

See Table 1

**Table 1.  Reagents and tools.**

| Reagent/resource | Reference or source<br>*Source (public): Stock center, company, other labs. Reference: list relevant study (e.g. Smith et al, 2019) if referring to previously published work; use "this study" if new. If neither applies: briefly explain.* | Identifier or catalog number<br><br>*Provide catalog numbers, stock numbers, database IDs or accession numbers, RRIDs or other relevant identifiers.* |
|---|---|---|
| **Experimental models** | | |
| L-Wnt3a cells from M. musculus | ATCC | CRL-2647 |
| B6;129P2 (M. Musculus) | Jax mice | B6;129P2-Or51e2tm1Mom/MomJ (referred as Olfr78-GFP) #006722 |
| B6, 129 (M. Musculus) | Jax mice | B6.Cg-Tg(Vil1-cre)997Gum/J (referred as Vil1Cre) #004586 |
| B6, 129 (M. Musculus) | Applied Stem Cell "this study" | B6-Olfr78Tm1Mig (referred as Olfr78Fx) |
| B6, 129 (M. Musculus) | Applied Stem Cell "this study" | B6-Olfr558Tm1Mig (referred as Olfr558-BFP) |
| M3x1 Olfr78-GFP WT mouse colon organoid line | This study | Passage at which experiments were done 13-19 |
| male M4x1 Olfr78-GFP WT mouse colon organoid line | This study | Passage at which experiments were done 13-19 |
| male M4x8 Olfr78-GFP WT mouse colon organoid line | This study | Passage at which experiments were done 13-19 |

**Table 1.** (continued)

| Reagent/resource | Reference or source<br><br>*Source (public): Stock center, company, other labs. Reference: list relevant study (e.g. Smith et al, 2019) if referring to previously published work; use "this study" if new. If neither applies: briefly explain.* | Identifier or catalog number<br><br><br><br>*Provide catalog numbers, stock numbers, database IDs or accession numbers, RRIDs or other relevant identifiers.* |
|---|---|---|
| male M10x8 Olfr78-GFP WT mouse colon organoid line | This study | Passage at which experiments were done 13-19 |
| male M3t2 WT mouse colon organoid line | This study | Passageat which experiments were done 3-9 |
| male M10t2 WT mouse colon organoid line | This study | Passage at which experiments were done 3-9 |
| male M10x4 Olfr78-GFP KO mouse colon organoid line | This study | Passage at which experiments were done 9-15 |
| male M11x4 Olfr78-GFP KO mouse colon organoid line | This study | Passage at which experiments were done 9-15 |
| male M13x4 Olfr78-GFP KO mouse colon organoid line | This study | Passage at which experiments were done 9-15 |
| male f1.17 Olfr78-GFP KO mouse colon organoid line | This study | passage at which experiments were done 9-15 |
| male M3x3 Olfr78-GFP KO mouse colon organoid line | This study | passage at which experiments were done 6-12 |
| male M4x3 Olfr78-GFP KO mouse colon organoid line | This study | passage at which experiments were done 6-12 |
| **Antibodies** | | |
| *example*: Rabbit anti-H3 | Abcam | Cat # ab1791 |
| Mouse anti-5HT | Dako | M0758, diluted 1/100 |
| Rabbit anti-BFP | Evrogen | AB233, diluted 1/500 |
| Mouse anti-PYY | Abcam | ab112474, diluted 1/200 |
| Rabbit anti-GFP | Invitrogen | A6455, diluted 1/500 |
| Rat anti-PE-Epcam | BD Biosciences | 563477, diluted 1/100 |
| Rabbit anti-KI67 | Abcam | ab15580, diluted 1/100 |
| Donkey anti-mouse-Cy3 | Jackson Immunoresearch | 715-165-150, dilution 1/500 |
| Donkey anti-rabbit-AF488 | Jackson Immunoresearch | 711-545-152, diluted 1/500 |
| Donkey anti-rabbit-biotinylated | Jackson Immunoresearch | 711-065-152, diluted 1/500 |
| Dapi | Sigma | D9542, diluted 1/2000 |
| **Oligonucleotides and sequence-based reagents** | | |
| PCR primers for Olfr78-GFP mice genotyping | Jax mice | For (CCTGTGATCAATCCCATCATC)/Rev (GGGTCTCATTTTACAGCAGAATC)/Mut For (CTACCATTACCAGTTGGTCTGGTG) |
| PCR primers for Vil1Cre genotyping | Jax mice | For (GTGTGGGACAGAACAAACC)/Rev (ACATCTTCAGGTTCTGCGGG) |
| PCR primers for Olfr78Fx genotyping | Applied Stem Cell "this study" | For (GAACTTTTCTCTCGTAGCTTGTACAGGA)/Rev (GACTGGAAGAGGGAGAGGCCAC) |
| PCR primers for Olfr558-BFP genotyping | Applied Stem Cell "this study" | For (CAGAATCTGACAATAACTTGGATGGTT)/ Rev (ATTCATTGCTATTGAAGCCCACC)/Mut For (CCTCTGATCTTGACGTTGTACATGAGG) |
| mouse Chgb | This study | For (CTAAGAGACCCAGCCGATGC)/Rev (CCTACCTTCGTCTGGCCTTG) |
| mouse Ffar2 | This study | For (TATGGAGTGATCGCTGCTCTG)/Rev (AGGTTATTTGGTTCTCAGTGCC) |

**Table 1.** (continued)

| Reagent/resource | Reference or source<br><br>*Source (public): Stock center, company, other labs. Reference: list relevant study (e.g. Smith et al, 2019) if referring to previously published work; use "this study" if new. If neither applies: briefly explain.* | Identifier or catalog number<br><br><br>*Provide catalog numbers, stock numbers, database IDs or accession numbers, RRIDs or other relevant identifiers.* |
|---|---|---|
| mouse Ffar3 | This study | For (TCCTCAGCACCCTCAACTCT)/Rev (TGGATGGCTCTTCTCCATTC) |
| mouse Gcg | This study | For (ACACCAAGAGGAACCGGAAC)/Rev (CCTGGCCCTCCAAGTAAGAAC) |
| mouse Olfr78 | This study | For (CGCTGCTGTCCTCAACAATA)/Rev (GTGGGAGAGCACATTGGAGT) |
| mouse Olfr558 | This study | For (CCTGCCTGTCATTATGGCTAAC)/Rev (TGGTCACGAGGAAAAGACGAA) |
| mouse Pyy | This study | For (TCAGTAGCTGTCGAGCCTTC)/Rev (ACAGGCGAGCAGGATTAGC) |
| mouse Rpl13 | This study | For (CCCGTGGCGATTGTGAA)/Rev (TCATTGTCCTTCTGTGCAGGTT) |
| mouse Tph1 | This study | For (CAAACTCTACCCGACCCACG)/Rev (CAGGACGGATGGAAAACCCA) |
| T. sanguinis | This study | For (CAGACGGGGACAACGATTGCA)/Rev (TACGCATCGTCGCCTTGGTA) |
| Universal 16 s rDNA | This study | For (GTGSTGCAYGGYYGTCGTCA)/Rev (ACGTCRTCCMCNCCTTCCTC) |
| mouse Ywhaz | This study | For (TGCAACGATCTACTGTCTCTT)/Rev (CGGTAGTAGTCACCCTTCATTTTCA) |
| mouse Olfr78 RNAscope probe | ACD-Biotechne | 436601 |
| mouse Olfr558 RNAscope probe | ACD-Biotechne | 316131 |
| mouse Ffar2 RNAscope probe | ACD-Biotechne | 433711 |
| mouse Ffar3 RNAscope probe | ACD-Biotechne | 447011 |
| **Chemicals, enzymes and other reagents** | | |
| 10% Formalin solution, buffered | Vwr | 11699408 |
| Sucrose | Millipore | 1076511000 |
| Tissue freezing medium | Leica | 14020108926 |
| Sodium acetate | Thermo Scientific Chemical | A13184 |
| Sodium propionate | Thermo Scientific Chemical | A17740 |
| Sodium citrate | AnalaR Normapur | 27833294 |
| Carmine red | Sigma-Aldrich | C1022 |
| Methylcellulose | Sigma-Aldrich | M0512 |
| Vectastain elite ABS kit, peroxidase | Vector Laboratories | PK-6100 |
| Vectainstain ABC kit- Alcaline phosphatase | Vector Laboratories | AK-5000 |
| DAB substrate kit | Vector Laboratories | SK-4100 |
| Vector blue substrate kit | Vector Laboratories | SK-5300 |
| Mayers' hemalun solution | Millipore | 1092492500 |
| Alcian blue 8GX | Sigma-Aldrich | A3157 |
| Nuclear fast red | Sigma-Aldrich | 229113 |
| Coverquick 4000 | VWR Chemicals | 5547539 |
| Rneasy mini kit | Qiagen | 74106 |
| Rneasy micro kit | Qiagen | 1071023 |

**Table 1.** (continued)

| Reagent/resource | Reference or source Source (public): Stock center, company, other labs. Reference: list relevant study (e.g. Smith et al, 2019) if referring to previously published work; use "this study" if new. If neither applies: briefly explain. | Identifier or catalog number Provide catalog numbers, stock numbers, database IDs or accession numbers, RRIDs or other relevant identifiers. |
|---|---|---|
| RNAse OUT | Invitrogen | 10777019 |
| Superscript II | Invitrogen | 18064-014 |
| DNAse I | Invitrogen | 18068-015 |
| RNAscope kit | ACD-Biotechne | 322300 |
| QIAzol Lysis Reagent | Qiagen | 79306 |
| Ovation Solo RNAseq systems | Tecan | 501 |
| Serotonin ELISA kit | Abnova | KA2518 |
| **Software** | | |
| GraphPad Prism 10 | https://www.graphpad.com | |
| qPCR4.0 | Analytik Jena | |
| qBase | Biogazelle | |
| GSEA MolSig | Broad Institute | |
| Degust | Monash Institute | |
| ZEN Blue 3.5 | Zeiss | |
| Seurat Package in R | Stuart et al, 2019 | |
| SCTransform | Hafemeister and Satija, 2019 | |
| Rstudio | / | |
| NDP.view2 | Hamamatsu | |
| Biorender | https://www.biorender.com/ | |
| **Other cell culture reagents** | | |
| | | Reference/Final concentration |
| Advanced-DMEM/F12 medium | Thermo Fisher Scientific | 12634028/1 X |
| DMEM | Gibco | 41965-039 (1 X) |
| Wnt3a-conditioned medium | ATCC L-Wnt3a CRL2647 | 50% |
| L-Glutamine | Thermo Fisher Scientific | 25030024 (2 mM) |
| N2 | Thermo Fischer Scientific | 17502048 (1 X) |
| B27 w/o vit.A | Thermo Fisher Scientific | 12587010 (1 X) |
| Amphotericin | Thermo Fisher Scientific | 152900026 (2.5 µg/ml) |
| Gentamycin | Thermo Fisher Scientific | 15750060 (40 µg/ml) |
| penicillin-streptomycin cocktail 100 X | Thermo Fisher Scientific | 15070063 (1 X) |
| UltraPure EDTA 0.5 mM, pH = 8 | Invitrogen | 15575-038 (5–20 mM) |
| HEPES | Thermo Fisher Scientific | 15630080 (10 mM) |
| N acetyl cysteine | Sigma-Aldrich | A9165 (1 mM) |
| mouse EGF | Peprotech | 315-09 (50 ng/ml) |
| mouse Noggin | Peprotech | 250-38 (100 ng/ml) |
| mouse Rspondin 1 | R&D Systems | 7150-RS-250 (100 ng/ml) |
| Nicotinamide | Sigma-Aldrich | N0636 (10 mM) |
| TryplExpress | Thermo Fisher Scientific | 12605028 (Ready-to-use) |

**Table 1.** (continued)

| Reagent/resource | Reference or source<br><br>*Source (public): Stock center, company, other labs.*<br>*Reference: list relevant study (e.g. Smith et al, 2019)*<br>*if referring to previously published work; use "this*<br>*study" if new. If neither applies: briefly explain.* | Identifier or catalog number<br><br><br><br>*Provide catalog numbers, stock numbers, database IDs or accession numbers,*<br>*RRIDs or other relevant identifiers.* |
|---|---|---|
| Trypsin 2.5% | Gibco | 15090-046 (0.25%) |
| DPBS | Thermo Fisher Scientific | 14190094 (1 X) |
| Basement membrane matrix, LDEV free Matrigel Y-27632 | Corning | 354234 (100%) |
| Y-27632 | Sigma-Aldrich | Y0503 (10 µM) |
| 40 µm cell strainer | Avantor/VWR | 7322757 |
| 70 µm cell strainer | VWR | 7322758 |
| Fetal bovine serum (FBS) | ThermoFisher | 10270106 |
| Collagenase I | Sigma-Aldrich | C9407 (0.5 mg/ml) |
| Propidium iodide / Acridine Orange Stain | Logos Biosystems | F23001 (10 times dilution) |
| Chromium Next GEM Single cell 3' Reagent kits 3.1 | 10X Genomics | / |
| **Other resources** | | |
| Nanozoomer digital scanner | Hamamatsu | |
| Axio Observer inverted microscope | Zeiss | |
| AE31 microscope/Moticam Pro camera | Motic | |
| qTower 3 | Analytik Jena | |
| FACS Aria I cytometer | BD Biosciences | |
| Fragment Analyzer 5200 | Agilent technologies | |
| Chromium controller-10X Genomics | 10X Genomics | |
| NovaSeq 6000 | Illumina | |
| iMark Microplate reader | Biorad | |

## Experimental animals

Animal procedures complied with the guidelines of the European Union and were approved by the local ethics committee (CEBEA from the faculty of Medicine, ULB) under the accepted protocols 720 N and 863 N. Mice were bred and maintained under a standard 12 h-light-dark cycle, with water and rodent chow ad libitum. Mice strains were B6;129P2-Or51e2tm1Mom/MomJ (referred as Olfr78-GFP), B6.Cg-Tg(Vil1-cre)997Gum/J (Vil1Cre) obtained from The Jackson Laboratory, B6-Olfr78Tm1Mig (referred as Olfr78Fx) and B6-Olfr558Tm1Mig (referred as Olfr558-BFP) generated by Applied StemCell. Primers used for mouse genotyping are listed in the Reagents and tools table (Table 1). Colon transit analysis was performed as reported by Koester et al (2021). Briefly, mice (males aged 2–6 months) were submitted to oral gavage (200 µl/25 g of body weight) with Carmine red 6%–0.5% methylcellulose at 9h00. The global gut transit time was defined as the time required to expel the first red fecal pellet. The number of pellets expelled during a period of 2 h was determined by placing mice in isolated cages. Collected pellets were individually weighted (a mean of 11 pellets were analyzed per mouse).

## Tissue processing and immunohistochemical analysis

Intestine samples were immediately fixed with 10% formalin solution, neutral buffered (Sigma-Aldrich) overnight at +4 °C and then sedimented through 20% and 30% sucrose solution sequentially (minimum 24 h each) before OCT (Leica) embedding. Histological and staining protocols as well as immunofluorescence/histochemistry experiments were performed on 6 µm sections. Sodium citrate 10 mM, pH 6 was used as epitope retrieval solution. The primary antibodies were incubated overnight at 4 °C. The secondary anti-species biotin- or fluorochrome-coupled antibodies were incubated 1 h at room temperature. The ABC kits and substrate Kits (all from Vector Laboratories) were used for immunohistochemistry revelation. Double immunohistochemistry was performed as recommended by manufacturer. DAPI or hematoxylin were used for nuclei staining. The primary and secondary antibodies used for staining are listed in the Reagents and tools table. For the Alcian Blue/Nuclear fast red staining, OCT sections were dried for 20 min at room temperature and washed 2 times in PBS for 5 min. Slides were incubated for 3 min in 3% acetic

acid and then for 20 min in Alcian blue solution (Sigma-Aldrich) at room temperature. Slides were rinsed in 3% acetic acid, running tap water for 2 min, followed by distilled water. They were then incubated for 3 min in Nuclear Fast Red (Sigma-Aldrich) at room temperature and rinsed in running tap water. Slides were mounted in a xylene-based medium (Coverquick 4000, VWR Chemicals) after dehydration or in FluorSave Reagent (Merck). Nanozoomer digital scanner (Hamamatsu) and Zeiss Axio Observer inverted microscope (immunofluorescence) were used for image acquisition. Quantification of epithelial SCFA receptors positive cells along the gut was performed on a minimum of 130 crypt/villus units per sample. Their localization within the colon crypts was quantified on a minimum of 30 cells for each receptor. Colocalization of GFP with 5-HT or PYY colon cells and colocalization of BFP with 5-HT or PYY cells were determined on a mean of $50 \pm 20$ cells and $36 \pm 14$ cells per sample, respectively. Quantification of Ki67$^{+ve}$ cells per crypt was performed on a mean of $27 \pm 5$ crypts per sample. Cell density for 5-HT/PYY/*Olfr558* or GFP$^{+ve}$ cells per mm$^2$ was analyzed on a mean surface of $1.17 \pm 0.15$ mm$^2$ of epithelium delimited by hand on NDP viewer. The number of biological samples (animals) used for each experiment is reported in Figures and Figure legends.

## Crypt isolation and ex vivo culture

The colon was cut in small pieces and put in 20 mM EDTA (Invitrogen) in DPBS (Gibco) for 30 min on ice and shaken at 80 rpm for crypts dissociation. A mechanical dissociation was then performed by ups-and-downs in a FBS pre-coated 10 ml pipette. The mix was passed through a 70 µm filter (Corning) and centrifuged at $300 \times g$ for 5 min. At this step, purified crypts were either collected for organoid culture or RNA seq. The organoid culture was performed as described (Sato et al, 2011). Briefly, the medium used consisted in Advanced-DMEM/F12 medium (Gibco) supplemented with 1X GlutaMAX (Gibco), N2 (Gibco) and B27 w/o vit.A (Gibco), gentamycin, penicillin-streptomycin cocktail, amphotericin, 10 mM HEPES (all from Thermofisher Scientific), 1 mM N-acetyl cysteine (Sigma-Aldrich), 50 ng/ml EGF, 100 ng/ml Rspondin 1 (R&D systems), 100 ng/ml noggin (both from Peprotech), and 10 mM nicotinamide (Sigma-Aldrich) and 50% Wnt3a conditioned medium produced by L Wnt-3A cells (ATCC CRL-2647) following manufacturer's instructions. Culture medium was changed every other day and after 8–9 days in culture, organoids were harvested and digested with TryplExpress (Thermo Fisher Scientific) for 5 min at 37 °C. Cells were further mechanically dissociated by ups-and-downs using a 200 µl tips and PBS was added (5-times TrypleExpress volume). Cells were centrifuged at 1300 rpm for 5 min and were (re)plated in Basement membrane matrix, LDEV free Matrigel (Corning). Culture media were supplemented with 10-µM Y-27632 (Sigma-Aldrich) in all initial seeding and replating experiments, for 48 h. For organoid stimulation experiments, organoid lines were replated and cultured for 7 days in the medium reported above (Sato et al, 2011). Then, medium was removed and replaced with DMEM (Gibco) in presence or absence of sodium acetate (20 mM final) or sodium propionate (10 mM final) for 48 h (with fresh medium change after 24 h). At day 9, organoids were collected for RNA extraction. Pictures were acquired with a Moticam Pro camera connected to Motic AE31 microscope.

## Gene expression analysis by qPCR and RNAscope

qRT-PCR was performed on total RNA extracted from adult mouse tissues or organoid cultures using the RNeasy Mini Kit (Qiagen). A DNAse I treatment (Invitrogen) was used to remove potential contaminant DNA. cDNA was prepared using Rnase-OUT and Superscript II according to the manufacturer's protocol (Invitrogen). qPCRs were performed on the qTower 3 from Analytik Jena. Gene expression levels were normalized to that of reference genes (Rpl13, Ywhaz) and quantified using the qBase Software (Biogazelle). Primer sequences are reported in the Reagents and tools table. In situ hybridization experiments were performed according to manufacturer instructions with the RNAscope kit (ACD-Biotechne) (probes listed in the Reagents and tools table).

## Serotonin measurements in fecal pellets

Stools were collected either at different time points during mouse lifetime or at sacrifice and stored at $-80$ °C. Fecal pellets were resuspended in the assay buffer of the Serotonin ELISA kit (Abnova) at 50 mg/ml. After 5 min of incubation at 37 °C, pellets were dissociated by vortexing and centrifuged at 3000 rpm for 3 min. The supernatant was collected and diluted 50 times to be used for the ELISA following manufacturer's instructions. Absorbance was determined at 450 nm using the iMark Microplate reader (BioRad).

## Fluorescence-activated cell sorting (FACS) of Olfr78-GFP$^{+ve}$ cells

Colon crypts from 4- to 7-month-old mice were isolated as described above and then resuspended in 4 ml of TryplExpress (ThermoFisher Scientific) for 15 min at 37 °C under agitation at 75 rpm. Ups-and-downs were performed with a P1000 pipette and incubated for a further 25 min at 37 °C and 75 rpm. After a second round of ups-and-downs, 8 ml of binding buffer [PBS-2 mM EDTA-2% BSA (Sigma-Aldrich) (w/v)] was added and the mix was passed through a 40 µm filter (Avantor). Cells were centrifuged at 1300 rpm for 10 min and the pellet was washed with 4 ml of binding buffer. After centrifugation at 1300 rpm for 3 min, cells were incubated with a Phycoerythrin-coupled anti-Epcam antibody (BD Biosciences) at 1/100 (v:v) in binding buffer for 30 min on ice. After 3 washes with binding buffer, cells were sorted using a FACS Aria I cytometer (BD Biosciences). FSC and SSC intensities were used for debris and doublets' exclusion. GFP$^{+ve}$ cells were identified using the FITC channel (Olfr78-GFP line) and Epcam through the PE channel. Epcam$^{+ve}$/GFP$^{+ve}$ cells were sorted from 2 pools of 2 HE (1028 and 2511 cells for each pool) and 2 KO mice (1123 and 1506 cells for each pool). Five thousand Epcam$^{+ve}$/GFP$^{-ve}$ cells were collected from a WT mouse as 2 individual samples. The cells were collected in QIAzol lysis reagent (Qiagen).

## Bulk RNA sequencing and Gene Set Enrichment Analysis (GSEA)

RNA was extracted using Rneasy mini kit or miRNeasy microkit (Qiagen) for crypts and sorted cells, respectively, following manufacturer's instructions, including the on-column DNAse step.

RNA quality was checked using a Fragment Analyzer 5200 (Agilent Technologies). Indexed cDNA libraries were obtained using the Ovation Solo RNAseq systems (Tecan) following manufacturer recommendations. The multiplexed libraries were loaded on a NovaSeq 6000 (Illumina) and sequences were produced using a 200 Cycles Kit. Paired-end reads were mapped against the mouse reference genome GRCm38 using STAR software to generate read alignments for each sample. The annotation files Mus_musculus.GRCm38.90.gtf was obtained from ftp.Ensembl.org. After transcripts assembly, gene level counts were obtained using HTSeq. For the RNAseq performed on purified colon crypts (Fig. 6), differentially expressed genes with minimum 2 CPM (count per million) in minimum 2 samples were identified with EdgeR method (FDR < 0.2, $\log_2$ fold change of 0.585) and further analyzed using GSEA MolSig (Broad Institute) (Subramanian et al, 2005). For the RNAseq performed on FACS-sorted cells (Fig. 2), differentially expressed genes with minimum 2 CPM in minimum 2 samples and $\log_2$ fold change >2 or <−2 were identified with EdgeR method (FDR < 0.001). For the RNAseq performed on FACS-sorted cells (Fig. 3), differentially expressed genes with more than 2 CPM in all the samples were analyzed as (KO mean − WT mean)/[(KO SD + WT SD)/2] is <−2 or >2 and log2-fold change >0.585 or <−0.585. GSEA analysis on these samples was performed using the identified DEGs. Heatmaps were generated using Heatmapper (Babicki et al, 2016).

### Single cell RNAseq (scRNAseq)

Colon tissues from one wild type and one Olfr78-GFP KO mice (respectively, 8 and 6 months old) were cut in small pieces and digested with 0.5 mg/ml Collagenase I (Sigma-Aldrich) for 25 min at 37 °C under agitation at 75 rpm. Ups-and-downs were performed with a 10 ml pipette and the samples were incubated for a further 20 min at 37 °C and 75 rpm. An equal volume of PBS-5 mM EDTA was added, and cells were centrifuged for 10 min at 50 g. Pelleted cells were digested with 5 ml of Trypsin 0.25% (Gibco) for 25 min at 37 °C and 75 rpm. Cells were pipetted up and down, passed through a 40 μm filter and pelleted at 50 g for 10 min. The pellet was washed 3 times with PBS-BSA 0.04% (w/v). Cell viability was evaluated by propidium iodide/Acridine Orange staining (85.9% and 90.3% viability in the WT and the KO, respectively) before processing through the Chromium Next GEM Single Cell 3' Reagent Kits V3.1 (10X Genomics) and sequenced on a Novaseq 6000 (Illumina).

Data were analyzed through the Seurat Package in R (Stuart et al, 2019) keeping only cells having between 1500 and 30,000 counts and less than 30% of genes coming from the mitochondrial genome. SCTransform was used as the scaling method (Hafemeister and Satija, 2019). The UMAPs were made using 20 dimensions and the clustering resolution was 0.3 for epithelial cells and 1 for enteroendocrine cells.

### Microbiota analysis

Mice were randomly allocated to cages at weaning. Stools were collected either at different time points during mouse lifetime or at sacrifice and stored at −80 °C. Metagenome analysis was performed by Eurofins Genomics and SCFA concentration quantification was performed by Creative Proteomics using gas chromatography. To analyze the relative prevalence of *Turicibacter sanguinis* by qPCR in the fecal microbiota, DNA was extracted from the stool after digestion with proteinase K (Sigma-Aldrich), followed by isopropanol and ethanol precipitation. 25 ng of DNA was used to perform qPCR to quantify the relative amount of *Turicibacter sanguinis* compared to universal 16 S rDNA levels.

### Statistical analysis

Statistical analyses were performed with Graph Pad Prism version 10. All experimental data are expressed as mean ± SEM. The significance of differences between groups was determined by appropriate parametric or non-parametric tests as described in Figure legends.

## Data availability

The datasets produced in this study are available at GSE229814.

## Peer review information

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

## Acknowledgements

We are grateful to Sumeet Singh Pal and Elif Sema Eski for helpful discussions on single-cell transcriptome analyses and Christine Dubois for FACS experiments. The project was funded by the Belgian "Région de Bruxelles-Capitale – Innoviris" (PHD 118 OLFRINGUT grant), the Funds "Fonds David et Alice Van Buuren", "Fondation Jaumotte-Demoulin" and "Fondation Héger-Masson", the non-for-profit organisation "Association Recherche Biomédicale et Diagnostic" and the Fond National de la Recherche Scientifique FNRS (CDR/J.0063.23F).

## Author contributions

**Gilles Dinsart**: Conceptualization; Data curation; Formal analysis; Funding acquisition; Investigation; Visualization; Methodology; Writing—original draft; Writing—review and editing. **Morgane Leprovots**: Formal analysis; Methodology; Writing—review and editing. **Anne Lefort**: Resources; Formal analysis; Methodology. **Frédérick Libert**: Resources; Software; Formal analysis; Methodology. **Yannick Quesnel**: Conceptualization; Supervision; Funding acquisition; Writing—review and editing. **Alex Veithen**: Conceptualization; Supervision; Funding acquisition; Methodology; Writing—original draft; Writing—review and editing. **Gilbert Vassart**: Funding acquisition; Writing—review and editing. **Sandra Huysseune**: Conceptualization; Supervision; Funding acquisition; Methodology; Writing—original draft; Writing—review and editing. **Marc Parmentier**: Conceptualization; Supervision; Funding acquisition; Methodology; Writing—original draft; Writing—review and editing. **Marie-Isabelle Garcia**: Conceptualization; Resources; Data curation; Software; Formal analysis; Supervision; Funding acquisition; Validation; Investigation; Visualization; Methodology; Writing—original draft; Project administration; Writing—review and editing.

## Disclosure and competing interests statement

The authors declare no competing interests.

# Expanded View Figures

**Figure EV1.  SCFA receptors exhibit unique expression profiles along the small intestine and colon.**

(**A**) Representative RNAscope pictures of mesenchymal expression of Olfr78 in the gut. (**B**) Representative RNAscope pictures of mesenchymal expression of Olfr558 in the colon. (**C**) Representative RNAscope pictures of Ffar3 expression in myenteric plexuses in the gut. (**D**) Representative RNAscope pictures of mesenchymal expression of Ffar2 in the Ileum. Data information: Scale bars: 100 µm (low views) or 25 µm (insets). Arrowheads identify isolated SCFA-expressing cells.

►

                                                

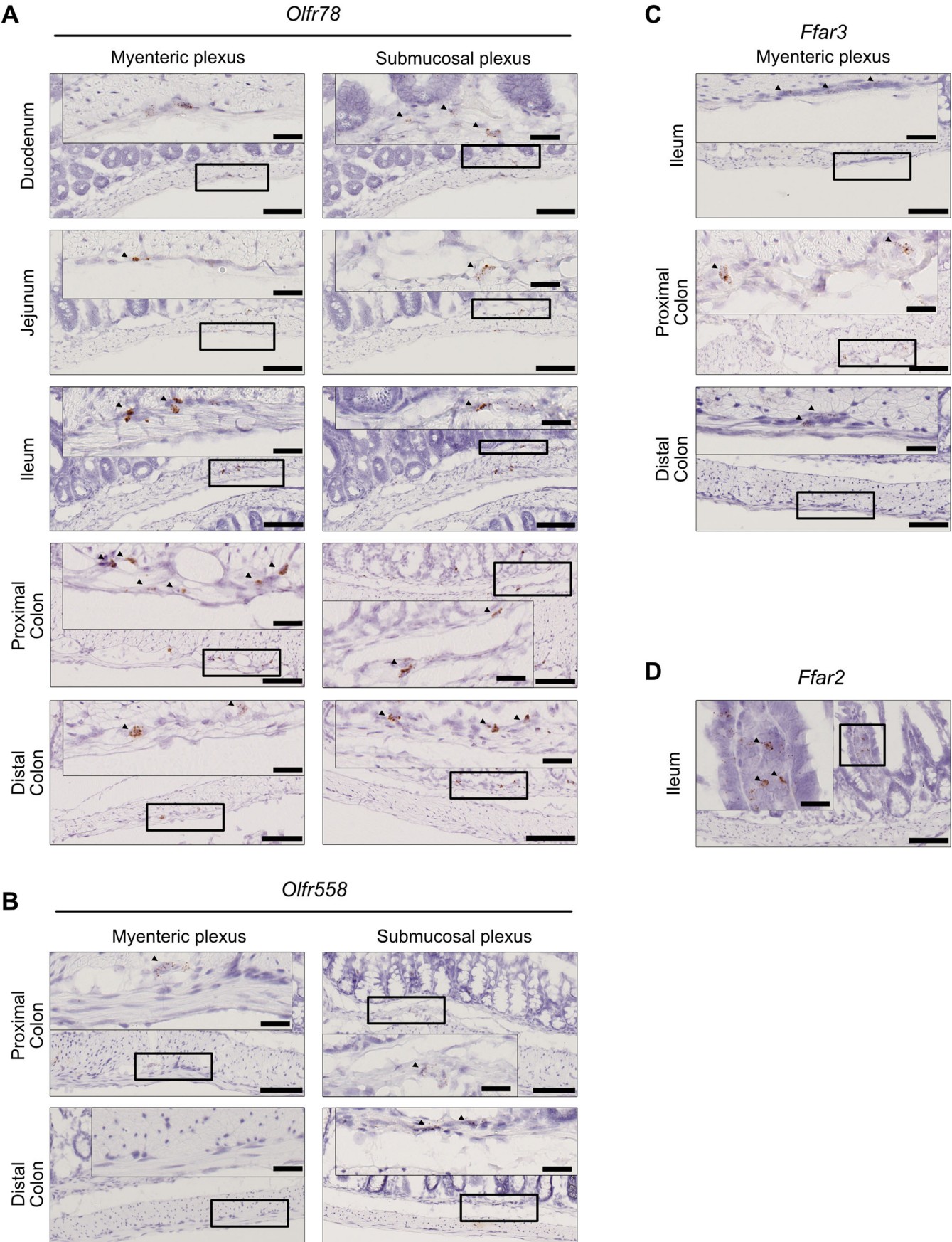

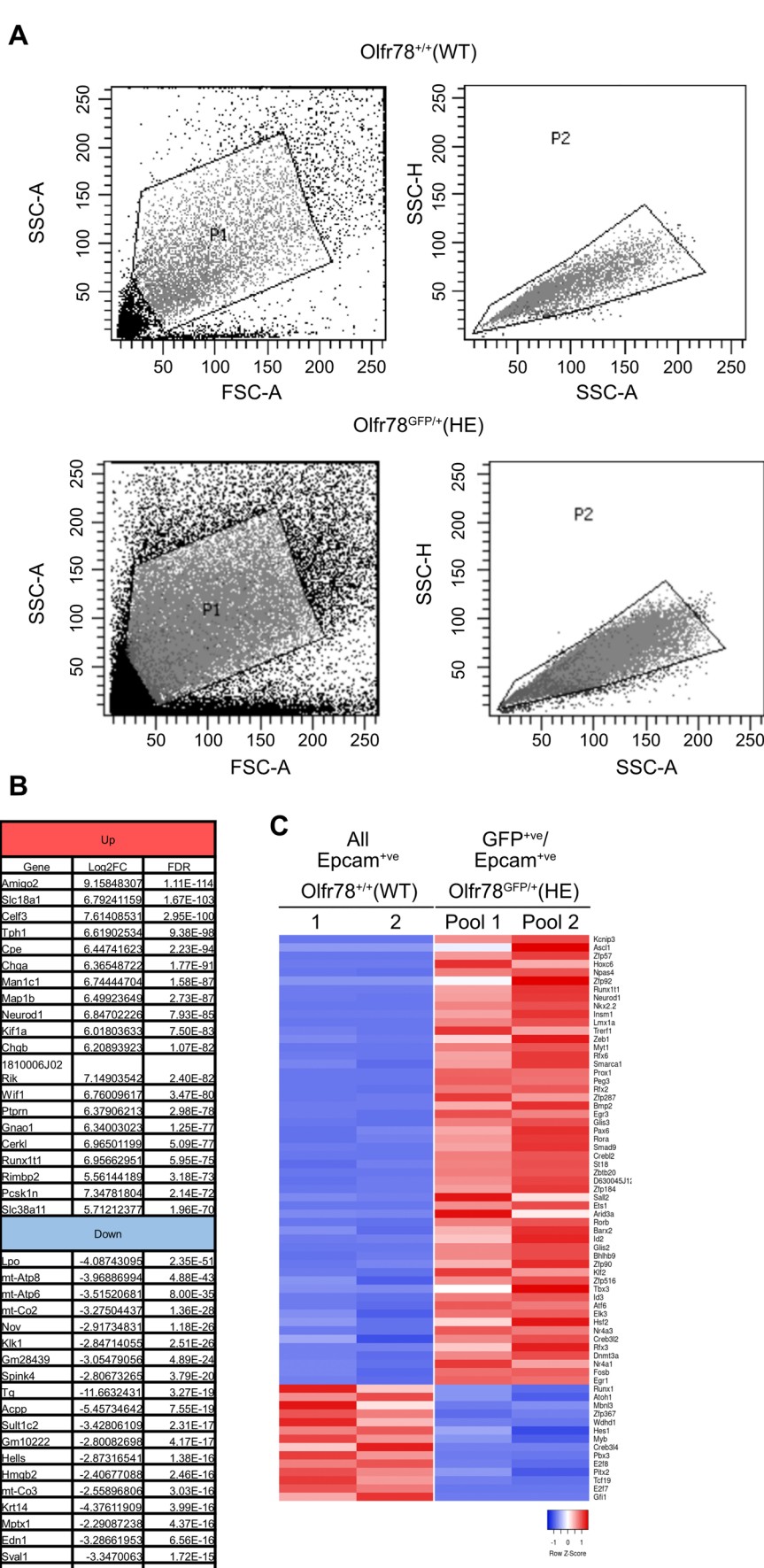

**Figure EV2.   Olfr78 is expressed in different subtypes of enteroendocrine cells in the colon.**

(A) FACS strategy for initial population selection and doublets exclusion. (B) List of the 20 most up or downregulated genes in Epcam$^{+ve}$/GFP$^{+ve}$ cells as compared to all Epcam$^{+ve}$ cells, ranked by FDR. (C) List of significantly up and downregulated transcription factors in Epcam$^{+ve}$/GFP$^{+ve}$ cells as compared to all Epcam$^{+ve}$ cells.

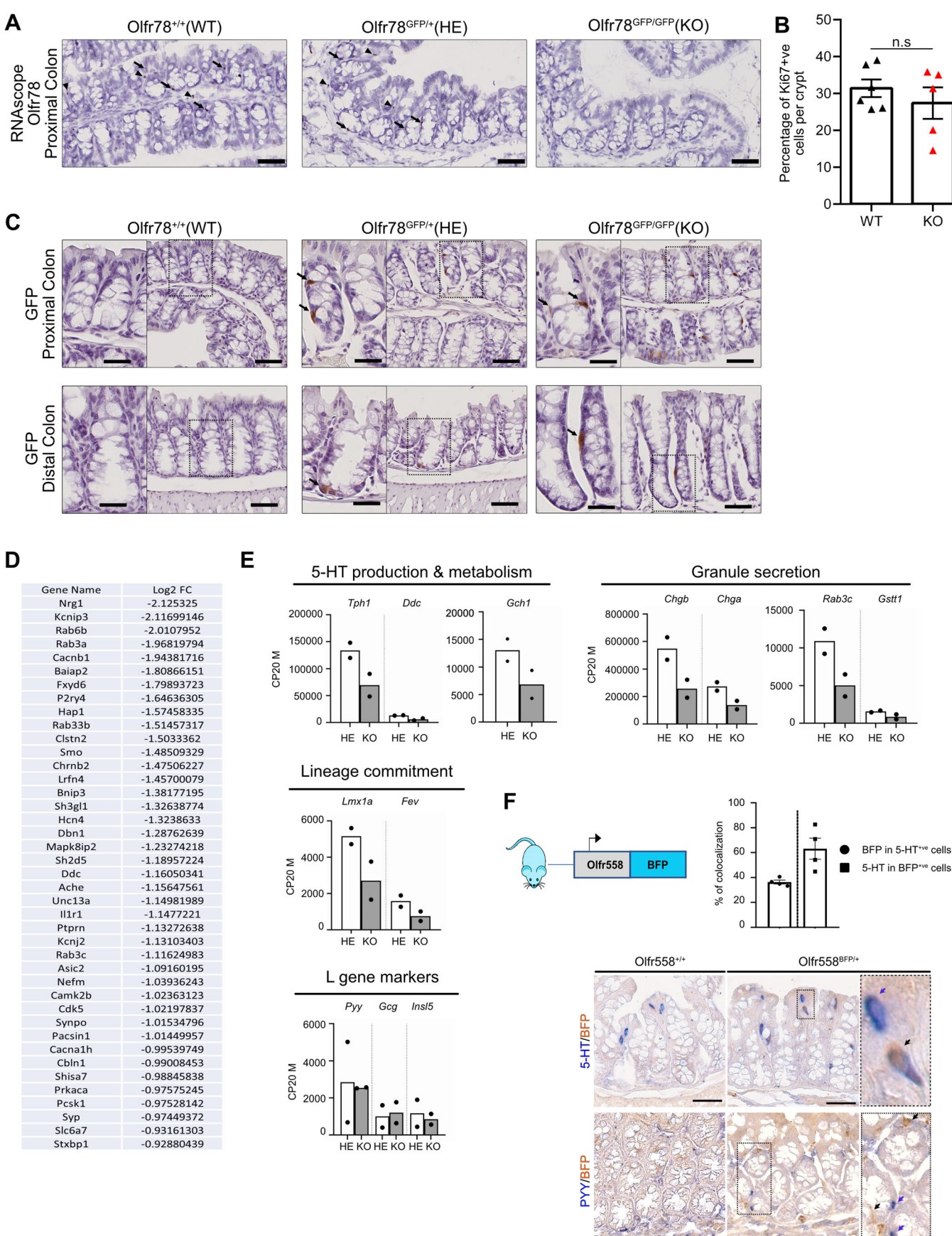

◀ **Figure EV3.  Loss of Olfr78 impairs terminal differentiation into enterochromaffin cells.**

(A) Olfr78 expression in proximal colon of WT, HE and Olfr78-GFP KO mice analyzed by RNAscope. Arrows and arrowheads identify epithelial and mesenchymal Olfr78-expressing cells, respectively. (B) Quantification of cell proliferation (Ki67$^{+ve}$ cells) in proximal colon crypts of Olfr78-GFP WT and KO mice. Each symbol indicates the value for a given mouse (n = 6 WT, 5 KO). (C) GFP expression in proximal colon of WT, HE and Olfr78-GFP KO mice analyzed by Immunohistochemistry. (D) List of downregulated genes in Olfr78-GFP KO Epcam$^{+ve}$/GFP$^{+ve}$ cells related to GSEA pre- or post-synapse gene lists, ranked by Log$_2$(Fold Change). (E) Histograms showing the expression of target genes in the bulk RNAseq from Fig. 3D. CP20M: counts per 20 million mapped reads. (F) Upper panel: Quantification of colocalization between 5-HT or PYY and BFP in the newly generated Olfr558-BFP mouse line (BFP cassette replacing the Olfr558 coding region). Each symbol indicates the value of a given mouse (n = 4). Lower panel: Representative pictures of double immunohistochemistry showing 5-HT/BFP and PYY/BFP co-stainings in proximal colon of Olfr558-BFP WT or HE mice. Data information: Scale bars: 50 μm (A, C and F) or 25 μm (A and C, inset). Data are represented as mean ± SEM.; unpaired t-test, n.s = not significant.

**A**

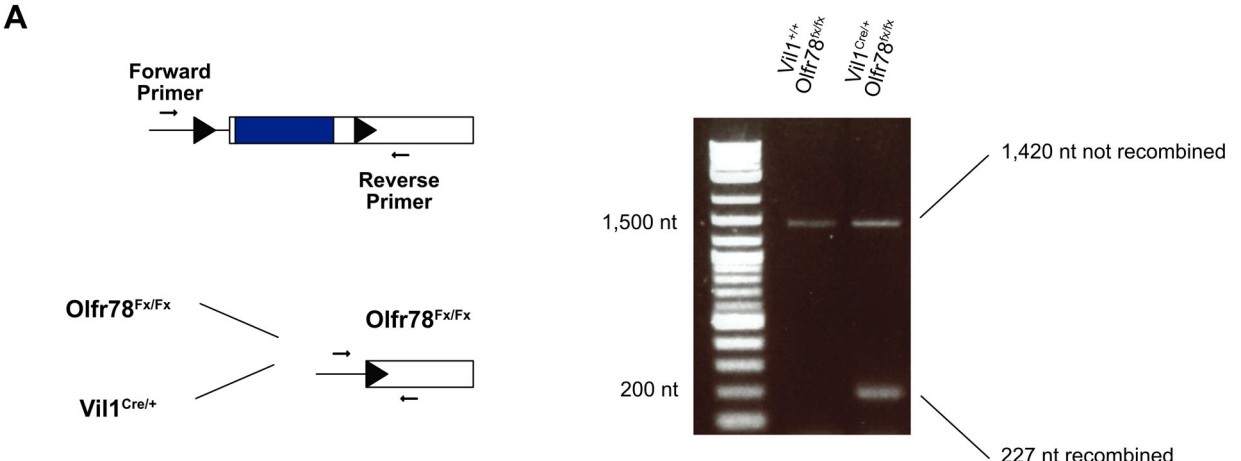

**B**

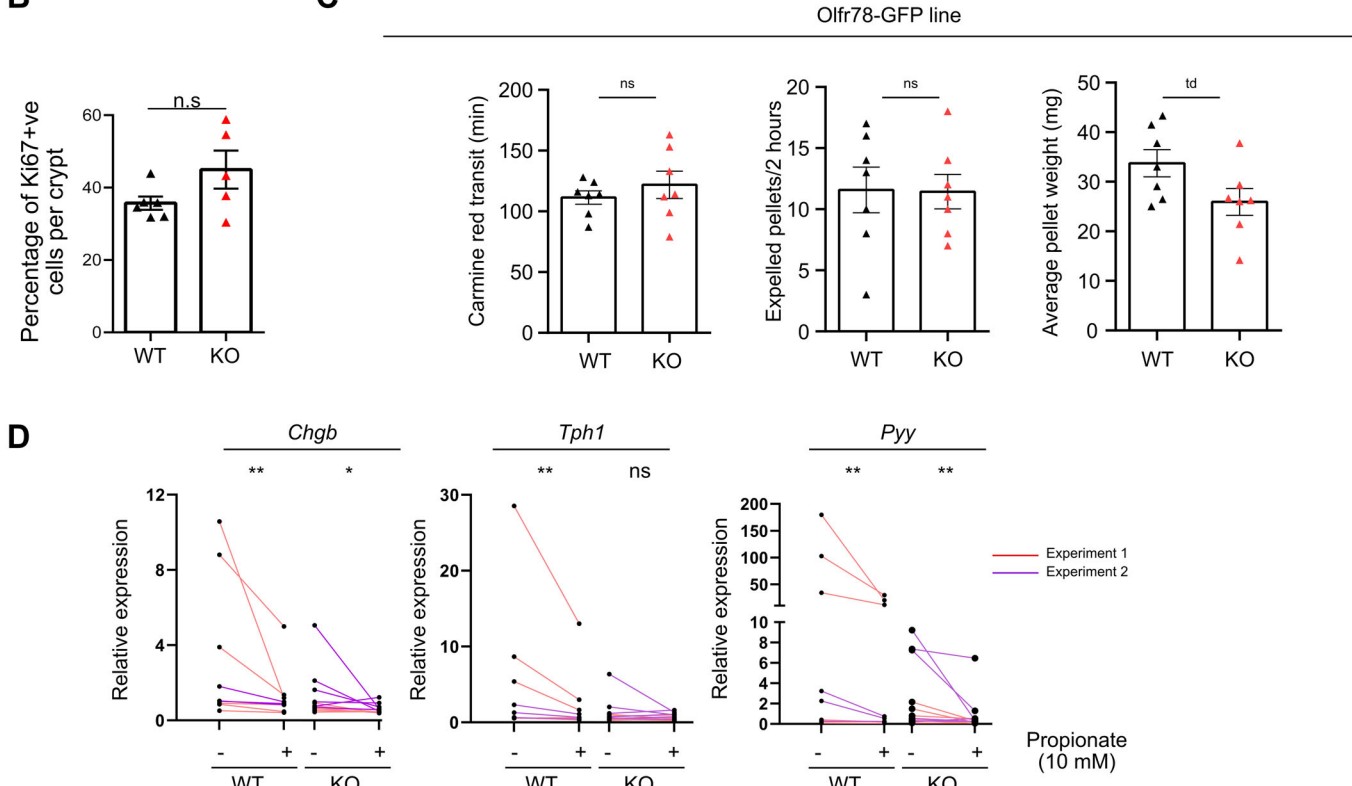

**E**

| Olfr558 Ct values | Mean ± sem (minimum-maximum) | |
|---|---|---|
| | Untreated (5) | Acetate 20 mM (5) |
| Olfr78+/+ | 30.55 ± 1.178 (26.38-32.53) | 30.20 ± 1.123 (25.87-32.11) |
| Olfr78GFP/GFP | 30.41 ± 1.242 (26.84-33.19) | 32.27 ± 0.139 (31.80-32.63) |

| Gcg Ct values | Mean ± sem (minimum-maximum) | |
|---|---|---|
| | Untreated (6) | Acetate 20 mM (6) |
| Olfr78+/+ | 30.05 ± 1.494 (26.6-35.0) | 30.05 ± 1.550 (25.2-33.8) |
| Olfr78GFP/GFP | 35.58 ± 0.971 (32.1-37.8) | 35.32 ± 0.661 (32.9-37.4) |

**Figure EV4. Terminal differentiation into serotonin-producing cells is regulated by epithelial Olfr78 expression.**

(A) Left: PCR strategy for loxp sites recombination verification in Vil1$^{Cre/+}$-Olfr78$^{fx/fx}$. Right: Gel electrophoresis showing WT and recombinant bands in Vil1$^{Cre/+}$-Olfr78$^{fx/fx}$. (B) Quantification of Ki67$^{+ve}$ cells in proximal colon crypts of Olfr78-GFP WT and KO mice. Each symbol indicates the value for a given mouse. (C) Histograms showing the transit time of carmine red (left), the number of fecal pellets expelled during 2 h (middle) and the average wet stools weight (right) in Olfr78-GFP WT or KO mice. Each symbol indicates the value for a given mouse ($n = 7$ WT and 7 KO). (D) Graphs showing the relative expression levels of EEC markers analyzed by qRT-PCR on colon organoids after 48 h of treatment with propionate at 10 mM. Data are reported as the relative expression levels in treated and control conditions in Olfr78-GFP WT and KO organoids. Each symbol indicates the individual value of a given organoid line in each experiment ($n = 3$–6 WT and 6 KO). Colored lines identify paired samples in 2 independent experiments. (E) Raw cycle threshold values obtained from qPCR experiments performed on Olfr78-GFP WT or KO organoids (mean ± SEM). Data information: Data are represented as mean ± SEM.; (B, C) unpaired t-test n.s = not significant; td: tendency ($p = 0.0673$), (D) Wilcoxon matched-pairs signed rank test, n.s = not significant, *$P < 0.05$, **$P < 0.01$.

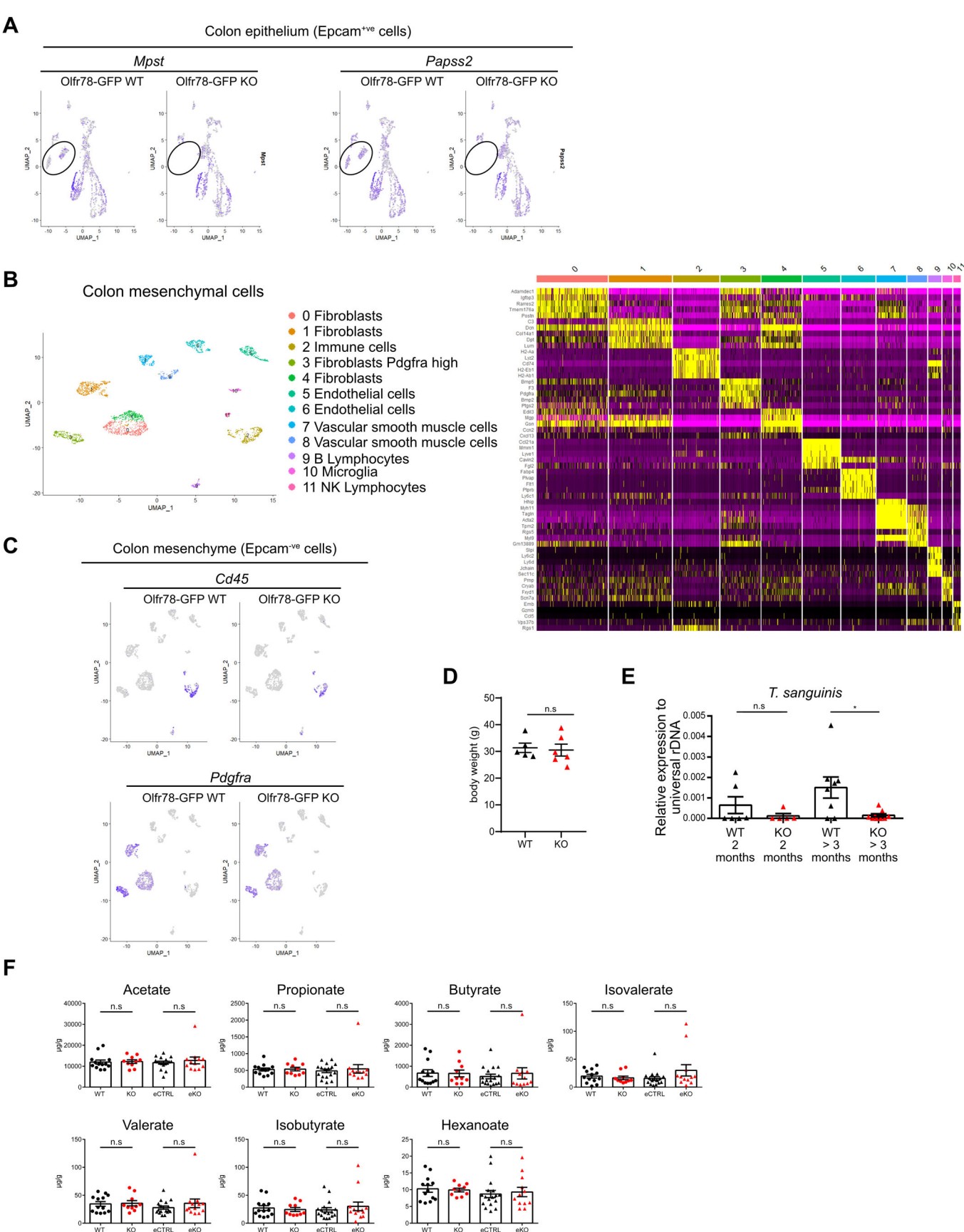

◀ **Figure EV5. Loss of Olfr78 expression alters colon homeostasis.**

(A) Umap of Olfr78-GFP WT and KO colon epithelial cells showing *Mpst or Papss2* expression. Circles identify enriched clusters differentially present in WT and KO mice. (B) Left panel: Merged UMAP of colon mesenchymal cells from Olfr78-GFP WT and Olfr78-GFP KO mice. Right panel: Heatmap showing the top 5 markers of each cluster of the UMAP. (C) Umap of Olfr78-GFP WT and KO colon mesenchymal cells showing *Cd45 or Pdgfra* expression. (D) Weight of adult Olfr78-GFP mice. Each symbol indicates the value for a given mouse ($n = 5$ WT and 6 KO). (E) Analysis of *Turicibacter sanguinis* prevalence by qPCR in the fecal microbiota of Olfr78-GFP WT or KO mice at different ages. Each symbol indicates the value for a given mouse ($n = 6$-8 WT and 5–9 KO). (F) Quantification of fecal SCFA concentrations. Each symbol indicates the value for a given mouse (Olfr78-GFP line: $n = 13$ WT and 10 KO; VilCre/Olfr78-Fx line: $n = 17$ controls and 12 eKO). Data information: Data are represented as mean ± SEM.; (D, E, F) Mann–Whitney tests, (E), n.s = not significant; *$P < 0.05$.

