## [Peer Review File · EMBO Reports]

THE OLFACTORY RECEPTOR Olfr78 PROMOTES DIFFERENTIATION OF ENTEROCHROMAFFIN CELLS IN THE MOUSE COLON

Marie-Isabelle Garcia, Gilles Dinsart, Morgane Leprovots, Anne Lefort, Frédéric Libert, Yannick Quesnel, Gilbert Vassart, Sandra Huysseune, Marc Parmentier, and Alex Veithen

DOI: [10.15252/embr.202357364](https://doi.org/10.15252/embr.202357364)

Corresponding authors: Marie-Isabelle Garcia (marie.garcia@ulb.be) , Gilles Dinsart (gilles.dinsart@ulb.be)

Review Timeline:

Submission Date:	19th Apr 23
Editorial Decision:	15th Jun 23
Revision Received:	12th Sep 23
Editorial Decision:	7th Nov 23
Revision Received:	10th Nov 23
Accepted:	15th Nov 23

Editor: Esther Schnapp

Transaction Report:

Dear Dr. Garcia,

Thank you for the submission of your manuscript to EMBO reports, and I am sorry for the unusual delay in getting back to you. We have now received the comments from 2 referees, which are pasted below.

As you will see, the referees acknowledge that the findings are potentially interesting. However, they also have several suggestions for how the study and the data should be strengthened, and I think that all should be addressed. I asked both referees for cross-comments but have not received a reply. Please let me know in case you have any comments or questions regarding the revisions, and we can discuss this further, also in a video chat, if you like.

I would thus like to invite you to revise your manuscript with the understanding that the referee concerns must be fully addressed and their suggestions taken on board. Please address all referee concerns in a complete point-by-point response. Acceptance of the manuscript will depend on a positive outcome of a second round of review. It is EMBO reports policy to allow a single round of major revision only and acceptance or rejection of the manuscript will therefore depend on the completeness of your responses included in the next, final version of the manuscript.

We realize that it is difficult to revise to a specific deadline. In the interest of protecting the conceptual advance provided by the work, we recommend a revision within 3 months (15th Sep 2023). Please discuss the revision progress ahead of this time with the editor if you require more time to complete the revisions.

- 1) A data availability section providing access to data deposited in public databases is missing. If you have not deposited any data, please add a sentence to the data availability section that explains that.
- 2) Your manuscript contains statistics and error bars based on $n=2$. Please use scatter blots in these cases. No statistics should be calculated if $n=2$.

5) a complete author checklist, which you can download from our author guidelines <https://www.embopress.org/page/journal/14693178/authorguide>. Please insert information in the checklist that is also

reflected in the manuscript. The completed author checklist will also be part of the RPF.

6) Please note that all corresponding authors are required to supply an ORCID ID for their name upon submission of a revised manuscript (<<https://orcid.org/>>). Please find instructions on how to link your ORCID ID to your account in our manuscript tracking system in our Author guidelines <<https://www.embopress.org/page/journal/14693178/authorguide#authorshipguidelines>>

7) Before submitting your revision, primary datasets produced in this study need to be deposited in an appropriate public database (see <https://www.embopress.org/page/journal/14693178/authorguide#datadeposition>). Please remember to provide a reviewer password if the datasets are not yet public. The accession numbers and database should be listed in a formal "Data Availability" section placed after Materials & Method (see also <https://www.embopress.org/page/journal/14693178/authorguide#datadeposition>). Please note that the Data Availability Section is restricted to new primary data that are part of this study. * Note - All links should resolve to a page where the data can be accessed. *
If your study has not produced novel datasets, please mention this fact in the Data Availability Section.

I look forward to seeing a revised form of your manuscript when it is ready. Please use this link to submit your revision:
<https://embor.msubmit.net/cgi-bin/main.plex>

Yours sincerely,

Referee #1:

In this manuscript, Dinsart and colleagues investigate the role of the short chain fatty acid (SCFA) receptor, Olfr78, in the colon using genetic and histological techniques. They first survey intestinal expression of three SCFA receptors and observe that Olfr78 is specifically expressed in the colon. They go on to characterize Olfr78-expressing epithelial cells via bulk RNAseq and histology in a reporter mouse line and, in agreement with previous reports, identify them primarily as enterochromaffin (EC) and L cells.

Subsequent single cell sequencing of epithelial cells from Olfr78-KO mice indicate a significant reduction in several signature EC cell genes, while L cell gene expression is unaffected. Some of these observations are supported by histological analysis and qPCR using colon tissue and organoids. Additionally, bulk RNA sequencing of the entire colonic epithelium from these mice show ontological differences between WT and KO mice. Minor differences in fecal microbiome are also reported.

A second genetic strategy is also used to eliminate Olfr78 expression specifically in the intestinal epithelium (using a villain-cre driver). These mice recapitulate the decreases in ChgB and Olfr558 expression and numbers of proximal colon 5HT+ cells observed in the global knockout.

Overall, the authors do provide some intriguing evidence that EC cell differentiation may be regulated by cell autonomous expression of Olfr78, including a decrease in the number of 5HT positive cells and reduction of EC cell gene expression in Olfr78 KO mice. However, there are several concerns:

-In figures 3 and 4, qPCR is used to support the scRNA sequencing results showing reduction in EC cell genes. Whole colon tissue samples are used for these experiments rather than dissociated epithelium, which would more directly sample the population of interest. ChgB, while largely specific to enterochromaffin cells within the intestinal epithelium, is also expressed in some neuronal populations and the authors describe expression of Olfr558 in non-epithelial cells. Decreased tissue expression in these genes cannot necessarily be attributed to EC cells. Further, the almost complete loss of Olfr558 in the global KO mice is inconsistent with the modest decrease in EC cell number. Of note, Tph1, which is a selective marker for EC cells in the colon, is not significantly affected in either mouse model, or are levels of 5HT in the epithelial-specific Olfr78 KO (for example, see Fig. 4g).

-The experiment in which ChgB expression is assessed following treatment of colonoids with acetate offers some support for the hypothesis that Olfr78 activation helps specify EC cell fate in a cell autonomous manner. Analysis of other genes such as Olfr558 and Tph1 would further support this idea, as would quantification of the number of 5HT-positive cells per organoid. Data from untreated organoids should be included as individual data points with appropriate statistical tests.

-The alterations in epithelial gene expression and microbiome composition described in Fig. 5 are not particularly compelling, and it is unclear how these experiments relate to the main hypothesis of the manuscript considering they are performed with global Olfr78 KO mice. Performing these experiments in Vill-Cre/Olfr78 fl/fl mice could more directly implicate EC cells and add significance to manuscript overall.

Referee #2:

In the present paper entitled "The Olfactory Receptor Olfr78 regulates differentiation of enterochromaffin cells in the mouse colon" (EMBOR-2023-57364V1) Dinsart and coworkers initially use QPCR and RNAscope in situ hybridization in WT mice but subsequently mainly study a knockout/knockin Olfr78 reporter mouse to characterize the expression of short chain fatty acid (SCFA) receptors with focus expression and function of OLF78. They find that Olfr78 and the closely related Olfr558 in contrast to the two ordinary SCFA receptors Ffar2 and Ffar3 mainly are expressed in 5HT enterochromaffin (EC) cells of the proximal colon and peptide hormone producing enteroendocrine (EEC) so-called L cells of the distal colon. Focusing on the Olfr78 expressing EC cells their transcriptomics analysis of the KO/KI mice indicates that loss of Olfr78 impairs terminal differentiation of the EC

cells. The authors also generate a more classical Villin-Cre driven Olf78 KO mouse line to further analyze this phenomenon including effects on general colonic gene expression and analysis of colonic organoids. Finally, they find alterations in the composition of the gut microbiota in Olf78 KO versus WT mice.

This is an impressive, well-executed study highlighting the importance of SCFA olfactory receptors for the function of colonic EEC and in particular EC cells which surprisingly identifies the involvement of Olf78 in the terminal differentiation of EC cells.

Major points:

1. This reviewer is confident that the Olf78 knockin/knockout reporter mouse in fact is a double knockout of Olf78 and Olf558. This is clear from the fact that expression of Olf558 with a very small variance is reduced to approx. 10-15% in the so-called Olf78 knockin/knockout mice (Fig. 3F and G), importantly, other EC cell genes e.g. Chgb is reduced on average by 20% (down to 80%) and with a huge variance probably as a consequence of the KO of Olf78 (and Olf558!) (Fig 3F and G). The problem is that the Olf558 gene is located VERY close to the Olf78 gene on the chromosome (gene duplication event), which also is the case for the two corresponding human genes, which are called OR51E1 and OR51E2, respectively. In fact, such a phenomenon of unintended double KO has been observed for other gene duplicated, closely located receptor genes. It would appear that this double KO in fact also has been realized by the authors and that they consequently made the villin-Cre driven Olf78 KO mouse to be sure. In that KO model the downregulation of Olf558 is more 'natural', i.e. around 50% with a large variance - similar to Chgb (Fig. 4E). Instead of 'hiding' this, the authors should 'admit' it and discuss the fact that these genes are so closely related and located. It should also be discussed to what degree the knockin reporter is a pure Olf78 reporter or a combined OLF78/Olf558 reporter?.

2. The authors identify some key changes in the gut microbiota in the 'real' OLF78 KO mice versus WT littermates and discuss the possibility that this is caused by changes in luminal secretion of 5HT being used by certain bacterial species. However, one of the major factors affecting gut microbiota, being even more important than e.g. obesity and diabetes etc., is colonic transit time (Roager et al. Nat Microbiol, 2016, 1:16093). As 5HT in the submucosa is a major driver of gut motility, the authors should take this into account and should e.g. included data on colonic the effect of Olf78 deficiency on colonic transit time if they included data on gut microbiota in this study where effects on 5HT producing cells is the major finding.

3. The stimulations of colonic organoids with SCFAs are problematic. The authors use 20 mM acetate to stimulate their organoid cultures (line 205-6 and Fig. 5H). This concentration is VERY high in particular for acetate and is likely damaging the cells. Along the acidity of this solution is problematic. Why not use propionate which is more potent on OLF78 and consequently could be administered in lower concentrations and even at the high concentrations is not as harmful for the cells as acetate? Stimulation for 48 hours is also very long. All and all these experiments are problematic and not very convincing. Please redo with the prototype SCFA ligand, propionate.

A few minor points:

1. Line 65 - the referenced Fleicher et al paper in fact not only reports that Olf78 is expressed in EEC L cells, but specifically tells that Olf78 is Not expressed in 5HT EC cells as reported in the present paper. What could be the reason for this difference?
2. line 200-2002 - strange use of the word 'dosage'? probably it means that you measure 5HT in the stools? (Fig. 4G).
3. Concerning L-cell markers - It is a little strange that the authors only use Pyy as a marker for L-cells. They should (also) show effects of Olf78 KO on the prototype L-cell marker gene, Gcg (precursor for GLP-1) in e.g. Fig. 4E. Strangely Gcg is also not mentioned in the discussion, when marker genes are listed (line 261)?

Editor's comment

As you will see, the referees acknowledge that the findings are potentially interesting. However, they also have several suggestions for how the study and the data should be strengthened, and I think that all should be addressed. I would thus like to invite you to revise your manuscript with the understanding that the **referee concerns must be fully addressed and their suggestions taken on board**.

First, we would like to thank editor and referees for positive consideration of our manuscript. As further detailed in the cover letter, all points raised by referees have been considered and additional experiments have been performed to address the asked questions. The revised manuscript (ms2) integrates new data and has been reshaped in text and figures. Changes are indicated (pages and lines correspond to the pdf-converted text with marks).

Overall, the new experiments further sustain the conclusions of the initial version of the ms (ms1).

Referee #1:

In this manuscript, Dinsart and colleagues investigate the role of the short chain fatty acid (SCFA) receptor, Olfr78, in the colon using genetic and histological techniques. (...).

*Overall, the authors do provide some intriguing evidence that EC cell differentiation may be regulated by cell autonomous expression of Olfr78, including a decrease in the number of 5HT positive cells and reduction of EC cell gene expression in Olfr78 KO mice. However, **there are several concerns**:*

*1.-In figures 3 and 4, **qPCR is used to support the scRNA sequencing** results showing reduction in EC cell genes. Whole colon tissue samples are used for these experiments rather than dissociated epithelium, which would more directly sample the population of interest. ChgB, while largely specific to enterochromaffin cells within the intestinal epithelium, is also expressed in some neuronal populations and the authors describe expression of Olfr558 in non-epithelial cells. **Decreased tissue expression in these genes cannot necessarily be attributed to EC cells.***

First, we would like to emphasize the fact that the qPCR experiments done on whole colon tissues were performed to validate bulk RNA sequencing data (Fig. 3D and fig. 3E in ms1) rather than scRNAseq data. Secondly, the bulk RNAseq presented in Figure 3 has been performed on **sorted epithelial GFP-positive cells** (using Epcam as a general epithelial marker). Therefore, the heatmap showing expression of EC and L cell markers refers specifically to *Chgb* expression in epithelial (EC) cells and downregulation of other gene markers can only be attributed to EC cells. Moreover, these conclusions, drawn from bulk RNAseq data, are further strengthened by single cell RNA seq showing downregulation of EC markers on epithelial Chga-expressing cells (Fig. 3H, page 29, line 736 of ms1).

- However, to facilitate data interpretation, we have modified Figure 3 by adding the following headings "bulk RNA seq" and "epithelial GFP⁺ve cells" in panels 3D and 3E, respectively. In the text of ms2, we have specified that transcriptome was obtained from bulk RNAseq on epithelial cells (results section: page 7, line 16, and figure legends of Fig. 3D and Fig. 3E, page 31, lines 6-10, respectively).

- Besides, as a complementary manner to evidence EC gene marker downregulation in Olfr78 KO GFP⁺ve cells in addition to the visual heatmap presented in Fig. 3E, we have added raw data from the bulk RNAseq for the following genes (*Tph1, Ddc, Gch1, Chgb, Chga, Gstt1, Lmx1a, Fev, Pyy, Gcg, Ins15*). The corresponding graphs, provided as new Fig. EV3E, show an average two-fold reduction in gene expression of EC, but not L, markers in Olfr78-deficient cells. This information has been introduced in the results section (page 7, line 23).

Therefore, taken altogether (qPCR on whole tissue + bulk RNAseq on Epcam⁺ve/GFP⁺ve cells), our data indicate that ChgB, and other EC markers are specifically downregulated in EC cells in Olfr78-deficient cells.

2. Further, the almost complete loss of *Olf558* in the global KO mice is inconsistent with the modest decrease in EC cell number.

First, as quantified in Figure 1C (2 *Olf558*⁺ cells per 100 crypts) and Figure 3G (20 *Olf558*⁺ cells per mm²) of ms1, the density of *Olf558* cells in proximal colon is quite low as compared to that of *Olf78* (10 cells per 100 crypts-Fig. 1C) and serotonin-producing cells (100 5HT⁺ cells per mm²). Then, to determine which proportion of epithelial 5HT⁺ cells do express *Olf558* and the other way around, we have used a newly generated mouse transgenic reporter line in which the BFP protein is expressed under the control of the *Olf558* promoter (referred as B6-*Olf558*Tm1Mig or *Olf558*-BFP). After having validated the reporter line, we have then performed co-staining for 5-HT and BFP expression in the proximal colon of adult heterozygous mice. Results of these quantifications are now added in the new Figure EV3F. Only a fraction (35%) of serotonin-expressing cells expresses *Olf558* whereas 60% of *Olf558*⁺ cells are 5-HT⁺. Altogether, these data reveal heterogeneity among 5-HT-expressing cells regarding odorant receptor co-expression: approximately 40% and 65% of 5-HT⁺ cells are *Olf78*-negative or *Olf558*-negative, respectively; this explaining the modest reduction in EC cells in absence of *Olf78*. In contrast, coherent with an almost complete loss of *Olf558* in *Olf78*-deficient mice, most *Olf558*⁺ cells would co-express *Olf78*. Therefore, we do not see manifest inconsistency in our findings. We have introduced quantifications on *Olf558*/5-HT cells in the new Figure EV3F. In ms2, the Results (page 8, lines 7-12), Material & Methods (page 18, line 5 and page 19, lines 8-9), Figure legends sections (page 35, lines 9-13) as well as Table 1 have been modified accordingly.

3. Of note, *Tph1*, which is a selective marker for EC cells in the colon, is not significantly affected in either mouse model,

We agree with referee 1 that expression of *Tph1* does not appear significantly modulated in *Olf78* KO mice when analyzed on whole tissues by qPCR (Fig. 3F and Fig. 4E). However, as discussed in the point 1 of referee's 1 comments (see above), isolated GFP⁺ KO cells demonstrate 2-fold reduced expression of *Tph1* as compared to GFP⁺ HE cells (see new Fig. EV3E). Moreover, we provide new analyses of *Tph1* expression in the epithelium and in EEC cells (*Chga*⁺ cells) by scRNAseq in the new Figure 3. These data show that few secretory progenitor cells do also express *Tph1* in the epithelium of *Olf78* WT and KO mice (Figure 3H) and that part of the undefined cluster of *Chga*⁺ cells in *Olf78*-GFP KO cells expresses *Tph1* as low levels (Figure 3I). We believe that this likely dampens the downregulation of *Tph1* expression in EC cells of *Olf78*-deficient mice at the whole tissue level. The Results (page 8 line 25 to page 9 line 1), Discussion (page 14, lines 19-21) and Figure legends (page 31, lines 19-23) sections have been modified to integrate these new findings.

4. or are levels of 5HT in the epithelial-specific *Olf78* KO (for example, see Fig. 4g).

As described in the Results section of ms1, measurement of 5-HT levels in stools (Figure 4G) indicated tendency to a 25% reduction in *Olf78*Fx cKO (n= 10) as compared to WTs (n = 16), though not reaching any statistical significance difference between genotypes. Such result is confirmed even when adding an additional KO mouse (p = 0.0797) in the new Figure 4G. We hypothesize that the presence of luminal 5-HT produced by the rest of "non-*Olf78*" EC cells present in the whole gut can compensate and mask the overall effect of *Olf78*-deficiency in the "*Olf78*"-EC cells. This is now discussed in ms2 (page 14, lines 14-16).

5. The experiment in which *ChgB* expression is assessed following treatment of colonoids with acetate offers some support for the hypothesis that *Olf78* activation helps specify EC cell fate in a cell autonomous manner. Analysis of other genes such as *Olf558* and *Tph1* would further support this idea, as would quantification of the number of 5HT-positive cells per organoid.

As detailed in point 6 just below, expression levels of *Tph1* have been added to the new Figure 4H. Data are now presented in a paired graph format showing individual points collected from 3 independent experiments using n=6 *Olf78* WT and n=6 *Olf78* KO organoid lines originating from 12 individual mice. Each independent experiment is represented with a different color line and each dot

represents an organoid line. Despite individual heterogeneity in cell differentiation, we have confirmed that acetate treatment increases EC (*Chgb*, *Tph1*) and not L (*Pyy*) gene marker expression in Olf78 WT organoid lines, a situation not observed in Olf78 KO lines, which overall demonstrate poor differentiation capacity as compared to WTs (see values of the y axes in the graphs of WT and KOs).

Regarding Olf78 and Gcg, with average Ct values around 30, these genes are expressed at low levels in organoid lines irrespective of the stimulation conditions, thus rendering qPCR data difficult to interpret. However, for complete transparency, we have introduced these data in the text (page 10, lines 17-18) and the new Figure EV4E as a table with mean of Ct values.

Regarding the asked quantification of number of 5-HT⁺ cells per organoid, first it is important to remind that *in vivo*, enteroendocrine cells and therefore EC cells, represent only a small proportion of whole epithelial cells (<1% in the colon). Secondly, *ex vivo*, colon organoids do not spontaneously differentiate over time as small intestine-derived organoids can do, due to the needed presence of Wnt3a to maintain stemness. By applying protocols reported to promote commitment towards the enteroendocrine lineage *ex vivo* on mouse colon organoids (Basak, Joep Beumer, Kay Wiebrands, et al. Induced Quiescence of Lgr5⁺ Stem Cells in Intestinal Organoids Enables Differentiation of Hormone-Producing Enteroendocrine Cells. *Cell Stem Cell*, 20, 177-190e4 (2017), <https://doi.org/10.1016/j.stem.2016.11.001>.) or human colon organoids (Zeve, D., Stas, E., de Sousa Casal, J. et al. Robust differentiation of human enteroendocrine cells from intestinal stem cells. *Nat Commun* 13, 261 (2022). <https://doi.org/10.1038/s41467-021-27901-5>), we have

previously faced the issue of getting EC differentiation associated with loss of the stem cell pool, a situation which does not satisfy “homeostatic conditions”. In the present study, organoid stimulation has been performed as follows: organoids have been cultured in Sato complete medium containing Wnt3a conditioned medium for 7 days after replating. Then, they have been stimulated for 2 days in DMEM alone in presence or not of ligands (acetate or propionate). We have now explicitly detailed the protocol in Material and Methods section (page 20, lines 3-7 of ms2). Such conditions are not massively inducing EC differentiation but preserve the Lgr5 stem cell pool. To answer referee’s question, we have performed 5-HT/DAPI staining on whole mount fixed treated/untreated organoids (see a picture with positive 5-HT cells in an untreated Olf78-GFP WT line on the right). However, we have been unable to faithfully quantify EC cells in organoids likely due to: 1) low proportion of 5-HT cells in organoids, 2) limited number of total organoids per sample (part of the well was also engaged in qPCR analyses), 3) heterogeneity among organoids in terms of differentiation ability (as also observed in qPCR studies, see point 6 below). Certainly, differentiation protocols should be further improved but for the present study, in such conditions, the more sensitive method of qPCR presented in Figure 4H allows to overcome these limitations and provides good evidence that Olf78, through acetate, promote maintenance of the EC phenotype.

6. Data from untreated organoids should be included as individual data points with appropriate statistical tests.

In ms1, data were represented as the gene expression fold-change in acetate versus untreated samples because of the high heterogeneity of differentiation we had observed in wells (discussed also in the point 5 above). However, we agree with referee 1 that this way of illustrating data loses global information. Therefore, as suggested, we have now presented qPCR results on organoid experiments as individual points showing data from 3 independent experiments (each experiment is visualized with a different color line, allowing to appreciate data reproducibility) in paired graph format in the new Figure 4H and Figure EV4D. Moreover, we have performed appropriate statistical tests, i.e. the Wilcoxon matched-pairs signed-rank tests. Gene expression is now reported for *Chgb*, *Tph1* and *Pyy*. As mentioned in the point 5, raw data Ct values for the lowly expressed Olf78 and Gcg genes are provided in Figure EV4E. For the Gcg gene, note that only 2 independent experiments are reported due to insufficient amount of cDNA left for one experiment. Several conclusions can be drawn from

these experiments. 1) individual organoid heterogeneity with overall higher capacity of WT organoids versus KOs to express EEC genes even at baseline (untreated conditions). 2) stimulation of expression of EC differentiation markers (Chgb and Tph1) but not Pyy in WT organoids upon acetate challenge, not observed in KO organoids.

7. -The alterations in epithelial gene expression and microbiome composition described in Fig. 5 are not particularly compelling, and it is unclear how these experiments relate to the main hypothesis of the manuscript considering they are performed with global *Olfr78* KO mice. Performing these experiments in *Vil-Cre/Olfr78 fl/fl* mice could more directly implicate EC cells and add significance to manuscript overall.

The analyses of the microbiome and whole colon epithelium of *Olfr78*-GFP mice were meant to investigate whether, even if modest, the altered EC differentiation detected in *Olfr78*-deficient mice could affect locally the rest of the epithelium and the flora. Indeed, it is reported that 5-HT released by the gut epithelium regulates gut bacterial composition (reviewed in : Mishima Y, Ishihara S. Enteric Microbiota-Mediated Serotonergic Signaling in Pathogenesis of Irritable Bowel Syndrome. International Journal of Molecular Sciences. 2021; 22(19):10235. <https://doi.org/10.3390/ijms221910235>).

However, knowing that *Olfr78* is also expressed in the stromal compartment and in other tissues outside the gut, as suggested by referee 1, we have now conducted the corresponding analyses for the *VilCre/Olfr78F_x* line with the following samples:

- bulk RNAseq on colon crypts: n = 3 control males (1 *Vil^{Cre/+}-Olfr78^{+/+}* and 2 *Vil^{+/+}-Olfr78^{F_x/F_x}*) and n = 2 eKO *Vil^{Cre/+}-Olfr78^{F_x/F_x}* males. Note that the number of samples involved in the study was limited by the available mice at the time of the revision of the ms.
- colon microbiome: n = 11 control males (5 *Vil^{Cre/+}-Olfr78^{+/+}* and 6 *Vil^{+/+}-Olfr78^{F_x/F_x}*) and n = 11 eKO *Vil^{Cre/+}-Olfr78^{F_x/F_x}* males.

Regarding the transcriptomic analysis, given the limited amount of available *Vil^{Cre}/Olfr78^{F_x}* samples, we have performed a grouped analysis with the *Olfr78*-GFP line leading to a total of n=5 *Olfr78* WT/controls versus n=7 *Olfr78* KO/eKO samples. As shown in the new heatmap (Figure 5A) and MDS plot (Figure 5B), loss of *Olfr78* leads to modulation of 75 genes (when applying FDR 0.2 and absolute log fold change of 0.585). Biological processes associated with the 40 downregulated genes indicated reduced potential for defense response against bacteria (new Figure 5C). Interestingly, expression of these genes (such as *Reg4*, *Saa1*, *Mpst*, *Papps2*) was detected in two clusters of Goblet-like populations in WT tissues (scRNAseq) that were missing in KO colon (Figure 5D and EV5A). Moreover, we detected an amplification of one cluster characterized as “secretory progenitors” in absence of *Olfr78* (Figure 5D). These findings further sustain the initial hypothesis that loss of this odorant receptor in EC cells alters cell fate in colon epithelium. The underlying mechanisms will require further investigation that is beyond the scope of the present work. These new data are introduced in Figure 5 of ms2 and the text: Abstract (page 2, line 9), Results (page 10 line 24 to page 11 line 8), Discussion (page 16, lines 15-19) and Material and methods (page 21, lines 21-23) sections are modified accordingly. The accession number of these new bulk RNAseq data is included in the GEOdata set GSE229814.

Regarding the microbiome of the *Vil^{Cre}-Olfr78^{F_x}* line, we have performed an analysis independent of the *Olfr78*-GFP line as major differences have been detected between the two mouse lines in terms of relative proportion and identity of the main Phyla (Figure 5F). Note that the altered *Firmicutes/Bacteroidetes* ratio described for *Olfr78*-GFP line in KO vs WT is not present in the other mouse line (new Figure 5G). In addition, the differences observed between *Olfr78* WT/controls and *Olfr78*/eCKO for genus and species are reported in Figure 5G. Altogether, these new findings suggest that loss of *Olfr78* is associated to slight alterations in the colonic flora.

These new data are introduced in the revised Figure 5. Text of the Results (page 11 line 16 to page 12 line 3) and Discussion (page 16, lines 1-3 and 9-15) sections are modified accordingly.

Referee #2:

This is an impressive, well-executed study highlighting the importance of SCFA olfactory receptors for the function of colonic EEC and in particular EC cells which surprisingly identifies the involvement of Olf78 in the terminal differentiation of EC cells.

We thank the referee for this positive feedback on our study.

Major points:

1. This reviewer is confident that the Olf78 knockin/knockout reporter mouse in fact is **a double knockout of Olf78 and Olf558**. This is clear from the fact that expression of Olf558 with a very small variance is reduced to approx. 10-15% in the so-called Olf78 knockin/knockout mice (Fig. 3F and G), importantly, other EC cell genes e.g. Chgb is reduced on average by 20% (down to 80%) and with a huge variance probably as a consequence of the KO of Olf78 (and Olf558!) (Fig 3F and G).

We totally agree with referee 2 that loss of Olf78 in the Olf78-GFP line leads to concomitant loss of Olf558 expression (reported in the former version in page 8, lines 158-160). A similar observation was made in the vilCre/Olf78Fx line in which residual 52% expression (page 9, lines 192-194 of the former ms) relies on the remaining mesenchymal expression of this odorant receptor.

1.1. The problem is that the Olf558 gene is located VERY close to the Olf78 gene on the chromosome (gene duplication event), which also is the case for the two corresponding human genes, which are called OR51E1 and OR51E2, respectively. In fact, such **a phenomenon of unintended double KO** has been observed for other gene duplicated, closely located receptor genes. It would appear that this double KO in fact also has been realized by the authors and that they consequently made the villin-Cre driven Olf78 KO mouse to be sure. In that KO model the downregulation of Olf558 is more 'natural', i.e. around 50% with a large variance - similar to Chgb (Fig. 4E).

Indeed, the potential generation of a double knockout phenotype due to the presence of these two genes in the same genetic location (distance of average 30 kb between Olf78 and Olf558) could be a possible reason for such phenotype. Note that in the Olf78-GFP mouse line, both epithelial and mesenchymal cells co-expressing both receptors are affected in the KO background. The observation that Olf558 was also strongly downregulated in the tissue-specific transgenic line, (albeit to a lesser extent since mesenchymal expression of Olf78 and Olf558 remains) led us to hypothesize that expression of Olf78 conditions that of the other odorant receptor. This was tentatively discussed in the Discussion section of ms1 (pages 13-14, lines 302-305).

1.2. Instead of **'hiding' this, the authors should 'admit' it** and discuss the fact that these genes are so closely related and located.

We respectfully disagree with this comment suggesting that we attempted to hide the fact that Olf78 KO mice are "double-KOs". Indeed, as explained in point 1.1, not only we described this phenotype in the Results section of the ms1 (page 8, lines 158-160); we also commented this unexpected finding in the Discussion section as well "Moreover, we unexpectedly found that the loss of Olf78 induced a decrease in Olf558 expression in EC cells. Although not formally excluded, the likelihood that this would result from an artifact of the transgenic strategy, affecting the genetic locus that bears the two genes, remains low since the effect was observed in both mouse lines" (pages 13-14, lines 302-305).

However, as suggested, we have now more explicitly described this notion in the Results section of ms2 in page 8, lines 3-5: "To our surprise, expression of the cognate SCFA receptor Olf558, whose gene lies 30 kb distant from the Olf78 gene on chromosome 17, dropped down to 19% residual levels in Olf78-deficient tissues (Figure 3F)" and page 9, lines 7-10: "Moreover, this study also aimed at determining whether the double Olf78/Olf558 KO phenotype observed in the Olf78-GFP line could result from unintended Olf558 KO phenotype due to the proximity of both genes on the genome".

1.3. It should also be discussed to what degree the knockin reporter is a pure Olf78 reporter or a combined OLF78/Olf558 reporter?

This is an important question, somehow related to point 1.2. In the epithelium, we have added additional data on the expression of Olf558 in proximal colon (not expressed in the epithelium of distal colon-Figure 1A,B and EV1B) in the Results section (page 8, lines 7-12) and the new Figure EV3F. For this purpose, we have used a new transgenic knockin mouse line B6-Olf558Tm1Mig (referred as Olf558-BFP) generated by Applied StemCell in which the BFP reporter is expressed under the control of the Olf558 promoter. Quantification of the number of cells coexpressing BFP and the EC (5-HT) or L (PYY) cell markers has revealed that only a fraction of EC cells (35%) coexpresses this odorant receptor meanwhile L cells do not coexpress Olf558 (Figure EV3F). Moreover, in contrast to the Olf78 receptor expressed in the epithelium of distal colon, Olf558 is not expressed at all in this part of the gut (only mesenchymal expression). Therefore, expression of both receptors does not fully overlap in epithelial colon, with expression of Olf78 being much wider than that of Olf558. This is further discussed in the Discussion section (page 15, lines 9-10).

The Materials and Methods section has been modified to introduce this mouse line and to provide the protocol for double immunohistochemistry (page 18, lines 8 and 23-25) as well as Table 1.

Further studies using a specific Olf558-deficient mouse line should help to clarify if expression of Olf558 depends on that of Olf78 during EC maturation and whether in turn, loss of Olf558 in epithelium or mesenchyme can also affect Olf78 expression.

2. The authors identify some key changes in the gut microbiota in the 'real' OLF78 KO mice versus WT littermates and discuss the possibility that this is caused by changes in luminal secretion of 5HT being used by certain bacterial species. However, one of the major factors affecting gut microbiota, being even more important than e.g. obesity and diabetes etc., is **colonic transit time** (Roager et al. *Nat Microbiol*, 2016, 1:16093). As 5HT in the submucosa is a major driver of gut motility, the authors should take this into account and **should e.g. included data on colonic the effect of Olf78 deficiency on colonic transit time** if they included data on gut microbiota in this study where effects on 5HT producing cells is the major finding.

As suggested by referee 2, we have now investigated colon transit time in our available Olf78-GFP mice (n = 7 WT and n = 7 KO mice). For this purpose, we have obtained a new ethic protocol (863N) reported in the Material and Methods section (page 18, lines 5 and 9-14). To investigate colon transit, we have used the protocol reported by Koester et al. (2021) which involves oral gavage with carmine red 6%. Transit time of carmine red throughout the gut (first observed expelled red pellet) expressed in minutes is reported in the new Figure EV4C. The average stool weight and number of stools released in 2 hours have also been studied. We didn't observe any statistical difference in colon transit time, number, or weight of stools between genotypes. As discussed in ms2 (page 14, lines 13-15), it is likely that the amount of serotonin released by Olf78⁺ve cells in the proximal colon only represents a small fraction of the total hormone released along the digestive tract; this explaining the absence of major impact on global colon transit.

Of note, we have now included the protocol for serotonin measurements in fecal pellets that was inadvertently omitted in ms1 (page 20, lines 18-24 of ms2).

3. The stimulations of colonic organoids with SCFAs are **problematic**. The authors use 20 mM acetate to stimulate their organoid cultures (line 205-6 and Fig. 5H). This concentration is VERY high in particular for acetate and is likely damaging the cells. Alone the acidity of this solution is problematic.

The choice of the concentration of acetate (20 mM) and propionate (10 mM) was based on the reported concentration of this metabolite in mouse colon that reaches 60 mM locally (Hernández MAG, Canfora EE, Jocken JWE, Blaak EE. The Short-Chain Fatty Acid Acetate in Body Weight Control and Insulin Sensitivity. *Nutrients*. 11, 1943. (2019). doi: 10.3390/nu11081943. Cong, J., Zhou, P., & Zhang, R. Intestinal Microbiota-Derived Short Chain Fatty Acids in Host Health and Disease. *Nutrients* 14, (2022). <https://doi.org/10.3390/nu14091977>).

As shown in the image of the medium used to stimulation organoids, the use of sodium acetate and sodium propionate allowed to maintain pH without any potential detrimental effect of culture medium acidification (phenol red is present in the DMEM medium as a pH indicator). Moreover, the representative pictures of the cultures 48 hours post-stimulation illustrate absence of major toxicity effect of the added ligands (see Figure 4H).

Therefore, we believe that the experiments performed at these concentrations are not problematic and are interpretable.

Why not use propionate which is more potent on OLF78 and consequently could be administered in lower concentrations and even at the high concentrations is not as harmful for the cells as acetate? Stimulation for 48 hours is also very long. All and all these experiments are problematic and not very convincing. Please redo with the prototype SCFA ligand, propionate.

We have now added results obtained with propionate stimulation at 10 mM in Figure 4H (for representative pictures of organoid cultures after stimulation) and Figure EV4D (for graphs). As shown in the picture above, the pH of the medium remained unchanged as well. Surprisingly, contrary to acetate, propionate treatment induced downregulation of EC and L cell markers in all organoids, potentially related to epigenetic modifications, based on published reports (Galoro da Silva et al., Sodium propionate and sodium butyrate effects on histone deacetylase (HDAC) activity, histone acetylation, and inflammatory gene expression in bovine mammary epithelial cells. *Journal of Animal Science* 96 (2019). 10.1093/jas/sky373; Grouls, M., et al. Differential gene expression in iPSC-derived human intestinal epithelial cell layers following exposure to two concentrations of butyrate, propionate and acetate. *Sci Rep* 12, 13988 (2022). <https://doi.org/10.1038/s41598-022-17296-8>).

Together, organoid experiments indicate that the two agonists do not exert similar effects on SCFA receptors. These new data are described in the Results section (page 10, lines 16-18) and discussed (page 14, lines 5-7). Table 1 has been modified accordingly.

Since EC cell differentiation takes few days (Gehart et al. Identification of Enteroendocrine Regulators by Real-Time Single-Cell Differentiation Mapping. 176, 1158-1173.e16. (2019). doi: 10.1016/j.cell.2018.12.029), we have chosen to investigate the impact of agonists on organoids after 48 hours to be able to detect expression of terminal differentiation EC or L cell markers. Since we didn't observe any sign of cell damage, we believe that 48 hours of stimulation to detect gene expression in relation to cell differentiation is a valuable timepoint.

A few minor points:

1. Line 65 - the referenced Fleicher et al paper in fact not only reports that Olf78 is expressed in EEC L cells, but specifically tells that Olf78 is Not expressed in 5HT EC cells as reported in the present paper. What could be the reason for this difference?

We agree with referee 2 on this point. One explanation would be that Fleischer et al. (2015) did not investigate specifically regional co-expression, in particular the proximal colon where coexpression of 5HT and the receptor is the highest. Indeed, we have found that a lower proportion of 5-HT+ve cells co-stain with GFP in the distal part, with higher variation between animals (see Fig. 2G). Importantly, our data are in line with other recent reports (as described in the text). However, as we do not have any proof for this hypothesis, we prefer not to comment in the text itself such discrepancy between the two studies.

2. line 200-2002 - strange use of the word 'dosage'? probably it means that you measure 5HT in the stools? (Fig. 4G).

We apologize for this mistake. We have corrected this sentence in ms2 (page 10, line 2) with “we measured this hormone in stools”.

3. Concerning L-cell markers - It is a little strange that the authors only use Pyy as a marker for L-cells. They should (also) show effects of Olf78 KO on the prototype L-cell marker gene, Gcg (precursor for GLP-1) in e.g. Fig. 4E. Strangely Gcg is also not mentioned in the discussion, when marker genes are listed (line 261)?

Gcg is a gene expressed in L cells but its expression is lower as compared to Pyy in Epcam⁺/⁺GFP⁺ HE cells (Figure EV3E).

However, as suggested by referee 2, we have now added qPCR data for this marker on whole colon tissues in both transgenic lines (Figure 3F and Figure 4E) in ms2. The text in the Results (page 8, line 2; page 9, line 24) and Discussion (page 13, line 11), sections as well as Table 1 (qPCR primers) have been modified accordingly.

Dear Dr. Garcia,

Thank you for your patience while your revised manuscript was re-reviewed. Unfortunately, referee 2 was not responsive, and I asked referee 1 to please also assess your response to referee 2's comments. I am happy to say that referee 1 supports the publication of your revised study, and we can therefore in principle accept it.

A few editorial requests still need to be addressed:

- Your ms has 5 main figures but the results and discussion sections are separate. Please either add one more main figure, or combine the results and discussion sections to publish your study as a short report with a maximum of 29,000 characters (including spaces but excluding references and materials and methods).
- Please correct the Conflict of Interest subheading to "Disclosure and Competing Interests Statement"
- Please remove the author credits from the ms file. All credits need to be entered now in our online ms submission system.
- Please correct the reference format to the EMBO reports style. "et al" needs to be used after 10 author names and DOIs should be used only for preprints and datasets that have not been published yet.
- The funding info Fondation Jaumotte-Demoulin and Fondation Héger-Masson are missing in our online ms submission system, please add this info when you upload your final ms files.
- The manuscript mentions Table 1 but the only table provided is called Reagent Table. Please correct the callout, the correct name of the table is "Reagent and Tools table" and there is a special file type in our online submission system for this table.
- The synopsis image looks good, but we need it in the correct, exact file size of 550 pixels wide x 200-400 pixels high. At this size, the text on the current image looks a little blurry. Please upload a new image at the correct size and using an image file (jpg, tif, etc).
- Please correct "EXTENDED FIGURES" to "Expanded View Figure Legends"
- In Figure 4B, please add highlight boxes for the zoomed figure panels.
- Please add the specific URL for the GSE229814 dataset in the data availability section.
- Please address our data editor comments in the final ms:
 1. Please indicate the statistical test used for data analysis in the legend of figure 5c.
 2. Please note that the box plots need to be defined in terms of minima, maxima, centre, bounds of box and whiskers, and percentile in the legend of figure 5g.
 3. Please note that information related to n is missing in the legend of figures 5g; EV5f.

I would like to suggest some minor changes to the title and abstract that needs to be written in present tense. Please let me know whether you agree with the following:

THE OLFACTORY RECEPTOR Olfr78 PROMOTES DIFFERENTIATION OF ENTEROCHROMAFFIN CELLS IN THE MOUSE COLON

The gastrointestinal epithelium constitutes a chemosensory system for microbiota-derived metabolites such as Short Chain Fatty Acids (SCFA). Here, we investigate the spatial distribution of Olfr78, one of the SCFA receptors, in the mouse intestine and study the transcriptome of colon enteroendocrine cells expressing Olfr78. The receptor is predominantly [OK?] detected in the enterochromaffin cells and L subtypes in the proximal and distal colon, respectively. Using the Olfr78-GFP and VilCre/Olfr78^{flox} transgenic mouse lines, we show that loss of epithelial Olfr78 results in impaired enterochromaffin cell differentiation, blocking cells in an undefined secretory lineage state. This is accompanied by a reduced defense response to bacteria in colon crypts and slight dysbiosis. Using organoid cultures, we further show that maintenance of enterochromaffin cells involves activation of the Olfr78 receptor via the SCFA ligand acetate. Taken together, our work provides evidence that Olfr78 contributes to colon homeostasis by promoting enterochromaffin cell differentiation.

I would also like to suggest some changes to the short summary and bullet points:

The short chain fatty acid receptor Olfr78 is expressed in enteroendocrine cells in the mouse colon where it promotes enterochromaffin differentiation through its ligand acetate.

- The olfactory receptor Olfr78, responding to SCFAs, is expressed in the enteroendocrine EC and L subtypes in mouse colon
- Loss of epithelial Olfr78 results in impaired EC terminal differentiation in the proximal colon
- Expression of the olfactory receptor Olfr558 is reduced in the colon of Olfr78-deficient mice
- Loss of Olfr78 expression alters colon homeostasis and is associated with slight dysbiosis

I look forward to seeing a final version of your manuscript as soon as possible. Please use this link to submit your revision:
<https://embor.msubmit.net/cgi-bin/main.plex>

Kind regards,
Esther

Referee #1:

The authors have addressed the initial criticisms with quite a bit of new data and various changes to the text. The study is still somewhat plagued by uncertainties stemming from EC cell heterogeneity and questions about whether disruption of the Olfr78 gene also affects a neighboring gene encoding Olfr558. However, the authors have now made these caveats more clear to the reader so that these issues can be taken into account. Several other technical issues are addressed, and the authors are correct in noting that some of these experiments are inherently difficult given the heterogeneity and paucity of EC cells in the gut epithelium and derived organoids. Overall, I believe that the revised manuscript is improved and should be considered for publication.

The authors addressed the minor editorial issues.

Dr. Marie-Isabelle Garcia
Universite Libre de Bruxelles ULB
IRIBHM
Route de Lennik 808
Brussels 1070
Belgium

Dear Dr. Garcia,

I am very pleased to accept your manuscript for publication in the next available issue of EMBO reports. Thank you for your contribution to our journal.
